# Phytochemicals for the Prevention and Treatment of Renal Cell Carcinoma: Preclinical and Clinical Evidence and Molecular Mechanisms

**DOI:** 10.3390/cancers14133278

**Published:** 2022-07-04

**Authors:** Essa M. Bajalia, Farah B. Azzouz, Danielle A. Chism, Derrek M. Giansiracusa, Carina G. Wong, Kristina N. Plaskett, Anupam Bishayee

**Affiliations:** College of Osteopathic Medicine, Lake Erie College of Osteopathic Medicine, Bradenton, FL 34211, USA; ebajalia81185@med.lecom.edu (E.M.B.); fazzouz65820@med.lecom.edu (F.B.A.); dchism04860@med.lecom.edu (D.A.C.); dgiansirac01847@med.lecom.edu (D.M.G.); cwong18138@med.lecom.edu (C.G.W.); kplaskett79131@med.lecom.edu (K.N.P.)

**Keywords:** renal cancer, phytochemicals, prevention, treatment, in vitro, in vivo, clinical studies, molecular mechanisms

## Abstract

**Simple Summary:**

Renal cell carcinoma (RCC) is the most frequently diagnosed kidney cancer. Once RCC metastasizes, successful treatment is difficult to achieve. There is an apparent need for novel approaches to prevent and treat RCC. Phytochemicals are naturally derived compounds gaining increasing scientific interest due to their cancer preventive and chemotherapeutic properties. These phytochemicals have been shown to exhibit a multitude of anticancer effects against RCC. In this systematic review, we critically evaluate the potential these natural compounds possess for the prevention and treatment of RCC and discuss the future implications this may have in the fight against kidney cancer.

**Abstract:**

Renal cell carcinoma (RCC) is associated with about 90% of renal malignancies, and its incidence is increasing globally. Plant-derived compounds have gained significant attention in the scientific community for their preventative and therapeutic effects on cancer. To evaluate the anticancer potential of phytocompounds for RCC, we compiled a comprehensive and systematic review of the available literature. Our work was conducted following the Preferred Reporting Items for Systematic Reviews and Meta-Analyses criteria. The literature search was performed using scholarly databases such as PubMed, Scopus, and ScienceDirect and keywords such as renal cell carcinoma, phytochemicals, cancer, tumor, proliferation, apoptosis, prevention, treatment, in vitro, in vivo, and clinical studies. Based on in vitro results, various phytochemicals, such as phenolics, terpenoids, alkaloids, and sulfur-containing compounds, suppressed cell viability, proliferation and growth, showed cytotoxic activity, inhibited invasion and migration, and enhanced the efficacy of chemotherapeutic drugs in RCC. In various animal tumor models, phytochemicals suppressed renal tumor growth, reduced tumor size, and hindered angiogenesis and metastasis. The relevant antineoplastic mechanisms involved upregulation of caspases, reduction in cyclin activity, induction of cell cycle arrest and apoptosis via modulation of a plethora of cell signaling pathways. Clinical studies demonstrated a reduced risk for the development of kidney cancer and enhancement of the efficacy of chemotherapeutic drugs. Both preclinical and clinical studies displayed significant promise of utilizing phytochemicals for the prevention and treatment of RCC. Further research, confirming the mechanisms and regulatory pathways, along with randomized controlled trials, are needed to establish the use of phytochemicals in clinical practice.

## 1. Introduction

Kidney cancer ranks as the seventh and tenth most common cancer in men and women, respectively, in the United States [1]. The American Cancer Society estimates that 79,000 new cases of kidney cancer will be diagnosed in 2022 and about 13,920 people will die from the disease [2]. Kidney cancer worldwide is estimated to account for 342,000 cases each year with more than 131,000 deaths [3]. Renal cell carcinoma (RCC) is the most common type of kidney cancer, with many subtypes according to histopathological findings, including clear cell (75%), papillary (10–15%), and chromophobe (5%), among others. Clear cell RCC cells display a lipid-rich cytoplasm, papillary RCC cells are organized into a spindle-shaped pattern with papillae, and chromophobe RCC cells are large and pale with perinuclear halos [4]. A unique characteristic of kidney cancer is that many patients are often asymptomatic; it is estimated that more than 50% of patients are diagnosed with kidney cancer incidentally during imaging for an unrelated issue [5]. Several major risk factors associated with kidney cancer include smoking, excess body weight, alcohol consumption, hypertension, diabetes, chronic kidney disease, as well as genetic factors, such as mutations in the von Hippel–Lindau (VHL) tumor suppressor gene [3].

Multiple signaling pathways have been identified for RCC. Hypoxic signaling is known to be mediated by hypoxia-inducible factors (HIF), which regulate the expression of target proteins, namely vascular endothelial growth factor (VEGF), platelet derived growth factor (PDGF), epidermal growth factor receptor (EGFR), and transforming growth factor-α (TGF-α). The VHL tumor suppressor gene serves as a ubiquitin ligase that targets HIF-α. In hypoxic RCC tumors (lacking VHL), HIF-α target proteins remain continuously expressed, leading to increased levels of VEGF, PDGF, EGFR, and TGF-α. VEGF, particularly in RCC, is strongly associated with the degree of angiogenesis, leading to increased tumor blood flow. Binding of VEGF and PDGF to their respective receptor tyrosine kinases activates phosphoinositide 3-kinase (PI3K), which ultimately activates Akt (also known as protein kinase B) via polycystin 1 and mammalian target of rapamycin (mTOR). This pathway has been reported to inhibit apoptosis in RCC cells. Wingless-related integration site (Wnt) signaling has been implicated in RCC through the induction of transcription by activating β-catenin. The hepatocyte growth factor (HGF) and receptor mesenchymal epithelial transition (MET) have been associated with papillary RCC. It has been proposed that the loss of VHL could lead to the HGF activation of β-catenin, permitting the development of RCC. Furthermore, inactivating point mutations in histone-modifying genes, such as set domain containing 2, jumonji AT-rich interactive domain 1C, ubiquitously transcribed tetratricopeptide repeat on X chromosome, and histone-lysine N-methyltransferase 2D, have been reported in 12–17% of clear cell RCC cases. The p16 and p14 tumor suppressor genes have been shown to be inactivated in 10–20% of primary RCC cases [6].

Localized RCC can be treated with partial or radical nephrectomy, depending on the size and location of the tumor. Robot-assisted partial nephrectomy is becoming a more popular option for the resection of localized tumors due to better preservation of renal parenchyma and ultimately renal function [7]. Other treatment options include ablation, radiation, cryotherapy, and active surveillance. The five-year survival rate for early diagnosed localized renal cancer is more than 90%, while the rate for distant metastatic disease is only 12% [8]. Metastatic RCC requires systemic therapy, which has not proved to be very effective. Chemotherapy and radiotherapy resistance pose a significant obstacle for the treatment of metastatic RCC [9]. Current treatment options for metastatic disease include tyrosine kinase inhibitors, such as sunitinib, pazopanib, cabozantinib, lenvatinib, and axitinib, and newer options, including checkpoint inhibitors nivolumab, ipilimumab, avelumab, and pembrolizumab [8]. Additional treatment options include bevacizumab, an anti-VEGF monoclonal antibody, and temsirolimus, an mTOR inhibitor [10]. Despite recent drug approvals for RCC, the need to identify new therapeutic options remains [11].

It is estimated that at least 20% of cancers can be prevented through dietary interventions, such as increasing the intake of fruits and vegetables [12]. Natural products, including bioactive phytochemicals, have gained significant attention in the scientific community and public for the prevention and treatment of various human cancers [13,14,15,16]. They have been utilized in the discovery and development of drugs, especially for cancer and infectious diseases [17,18,19]. Approximately 60% of current cancer chemotherapeutic drugs were derived directly or indirectly from natural sources [17]. Examples of the associations between natural products and anticancer agents include the vinca alkaloids (vincristine and vinblastine) isolated from the Madagascar periwinkle plant, taxanes (paclitaxel) derived from the bark of a Pacific yew tree, and camptothecins (irinotecan and topotecan) isolated from a Chinese ornamental tree [17]. Natural product-derived compounds, especially phytochemicals, have been shown to inhibit various stages of carcinogenesis based on preclinical and clinical studies by affecting the proliferation, cell cycle progression, cell death resistance, energy metabolism, cellular senescence immune response, epithelial to mesenchymal transition, angiogenesis, invasion, migration and metastasis of cancer cells [16,20,21,22,23,24,25]. Various mechanisms of these anticancer effects include upregulation of genes that detoxify reactive oxygen species (ROS) and main redox hemostasis, inhibition of inflammatory cascades, and modulation of numerous oncogenic and oncosuppressive signaling pathways [26,27,28,29,30,31,32,33].

In a previous review on natural therapy for RCC, the anticancer effects of various phytochemicals were analyzed. The review was limited to less than ten phytochemicals and did not address specific mechanisms of action [34]. In another publication, natural products were compiled into a review to examine the effects they have previously shown in RCC [35]. Numerous phytochemicals were identified, and their mechanisms of action were elaborated. However, the review was not completely comprehensive, and since then, numerous in vitro, in vivo, and clinical studies have identified additional phytochemicals that displayed anticancer effects in RCC. In an additional review, the phytochemical resveratrol was examined for its health benefit in kidney disease [36]. This study was limited to one phytocompound, but the results were not limited solely to renal cancer and also covered other renal ailments, such as chronic kidney disease and renal fibrosis. The aim of our study was to provide an updated and comprehensive evaluation of the phytochemicals to prevent or treat RCC, with a special emphasis on the specific anticancer mechanisms of action. With RCC being an aggressive and deadly cancer, there is great potential for the use of phytochemicals as drug therapy. A comprehensive review of phytochemicals and their anticancer effects in RCC will further consolidate information and serve as a reference for future researchers engaging in novel drug therapy. We also discuss the limitations to the use of phytochemicals and the future direction they have in the fight against kidney cancer. To our knowledge, this is the most up-do-date, critical, and comprehensive review of available research on RCC prevention and intervention utilizing bioactive phytochemicals.

## 2. Methodology for Literature Search and Study Selection

This study followed the recommended systematic review protocol, Preferred Reporting Items for Systematic Reviews and Meta-Analyses (PRISMA) criteria [37], for study selection as presented in Figure 1. This review has not been registered in the International Prospective Register of Systematic Reviews. Articles were collected using databases, primarily PubMed. No year constraints were placed on research articles. Keywords used during the search were phytochemicals, renal, kidney, cancer, phenolics, terpenoids, alkaloids, sulfur compounds, prevention, treatment, in vitro, in vivo, and clinical studies. Abstracts were reviewed to determine relevance and application to the study. Next, full-length articles were collected and evaluated for inclusion in or exclusion from the study. Only studies published in English were included, and abstracts and book chapters were excluded. Clinical studies were searched using clinicaltrials.gov. The literature search was independently conducted by six researchers (EMB., DAC., KNP, FBA, DMG, and CGW) and final reviews were conducted by a senior researcher (AB).

## 3. Phytocompounds in Renal Cancer Research: Preclinical Studies

### 3.1. Phenolics

#### 3.1.1. Alpinumisoflavone

Alpinumisoflavone (Figure 2) is a naturally occurring flavonoid found in the plant *Derris eriocarpa* F.C. as well as in various fruits, including mandarin melon berry (*Cudrania tricuspidate*). It has exhibited antineoplastic effects via suppression of proliferation, migration, and invasion as well as the promotion of apoptosis in various cancer models [38]. Alpinumisoflavone was investigated for its anticancer effect on human clear cell RCC cell lines 786-O, Caki-1, and RCC4. Alpinumisoflavone suppressed cell growth and invasion, and induced apoptosis in RCC4 and 786-O cells (Table 1). The ratio of phosphorylated Akt (p-Akt) to total Akt was decreased by treatment with alpinumisoflavone. The modulating activity of alpinumisoflavone on microRNA 101 (miR-101) and Ral interacting protein of 76 kDa (RLIP76) was reversed by Akt activation, suggesting that alpinumisoflavone inhibited Akt signaling, which was responsible for its anticancer effect [39]. Wang et al. [39] also evaluated the antitumor effect of alpinumisoflavone (40 or 80 mg/kg) in vivo against 786-O xenografts in BALB/c male mice, where it was shown to suppress tumor growth and metastasis (Table 2). Alpinumisoflavone treatment also led to an increase in apoptotic cells and miR-101 expression, while decreasing RLIP76 expression and the p-Akt/t-Akt ratio in tumor tissue.

#### 3.1.2. Angelicin

Angelicin (Figure 2), a furocoumarin, can be found in plants such as the Japanese medicinal plant known as Inutoki or Yamaninjin (*Angelica shikokiana*), wild angelica (*Angelica sylvestris* L. var. sylvestris), *Bituminaria basaltica*, and common fig (*Ficus carica* L.), and has exhibited antiviral, anti-inflammatory, and anticancer properties both in vitro and in vivo [136,137]. Min et al. [40] explored angelicin’s ability to enhance tumor necrosis factor-related apoptosis-inducing ligand (TRAIL)-induced cell death in human RCC Caki cells. Although angelicin alone failed to induce apoptotic cell death, a combination of angelicin and TRAIL elicited accumulation of a sub-G1 cell population and cleavage of poly (ADP-ribose) polymerase (PARP), producing apoptotic morphology. Additionally, treatment with this combination increased caspase-3 activation and downregulated cellular FLICE-like inhibitory protein (c-FLIP) expression, suggesting that angelicin potentiated TRAIL-induced apoptosis in Caki cells.

#### 3.1.3. Apigenin

Apigenin (Figure 2), a natural flavonoid present in parsley, celery, and chamomile plants, has shown antitumor activity both in vitro and in vivo, displaying modulatory effects in numerous cell signaling pathways [138,139]. To determine apigenin’s effects on renal cancer, human RCC lines ACHN, 786-O, and Caki-1 were used. Apigenin reduced proliferation in all three cell lines. In ACHN cells, apigenin induced DNA damage and showed an increase in H2A histone family member X phosphorylated on serine 139 (γH2AX), a marker for double-stranded DNA breaks. G2/M cell cycle arrest was observed in ACHN cells with an increase in phosphorylated-ataxia telangiectasia mutated (p-ATM), phosphorylated checkpoint kinase 2 (p-Chk2), phosphorylated Cdc25 on serine 216 (p-Cdc25c), and phosphorylated Cdc2 on tyrosine 15 (p-Cdc2) after treatment with apigenin. Expression of Bcl-2 associated X-protein (Bax), cleaved caspase-3, and cleaved caspase-9 all increased after apigenin treatment [41].

Meng et al. [41] also tested apigenin’s effect in vivo by treating BALB/c male mice with xenografted ACHN tumors with 30 mg/kg of apigenin, administered via intraperitoneal (i.p.) injection every 3 days for a duration of 21 days. This resulted in a reduction in tumor growth and volume. Tumors derived from the apigenin-treated mice were also shown to have reduced Ki-67 expression.

#### 3.1.4. Chrysin

Chrysin (Figure 2) is a naturally occurring flavone found in honey, propolis and flowering plants, such as the blue passionflower (*Passiflora caerulea*), the purple passionflower (*Passiflora incarnata*), and the Indian trumpet tree (*Oroxylum indicum*). Previously, it has exhibited antiviral, antioxidant, anti-inflammatory, and anticancer properties [140,141]. In an in vivo study using male Wistar rats, treatment with N-nitrosodiethylamine (DEN) and ferric nitrilotriacetate displayed massive inflammatory cell invasion with hyperchromatism, suggesting renal cancer formation. Oral administration of chrysin (20 or 40 mg/kg) was found to suppress DEN-induced renal carcinogenesis in rats. Mechanistically, chrysin supplementation increased glutathione *S*-transferase (GST) and quinone reductase (QR) levels and reduced proliferating cell nuclear antigen (PCNA)-positive cells in renal tissue. Furthermore, ornithine decarboxylase (ODC) activity was reduced, suggesting chrysin can inhibit cellular proliferation and tumor production. Additionally, chrysin was found to inhibit nuclear factor-κB (NF-κB) activation, reduce overexpression of cyclooxygenase-2 (COX-2) and inflammatory cytokines, such as interleukin 6 (IL-6), tumor necrosis factor-α (TNF-α), and prostaglandin E2 (PGE2), enforcing chrysin’s preventative effects against renal carcinogenesis [131].

#### 3.1.5. Coumarin

Coumarin (Figure 2) is a naturally occurring compound, highly concentrated in tonka beans (*Dipteryx odorata*), and can be found in vanilla grass (*Anthoxanthum odoratum*) and Chinese cinnamon (*Cinnamomum cassia*). It is also found in various fruits, such as strawberries, cherries, apricots, and black currents. Coumarin has been shown to exhibit anti-inflammatory and anticancer properties [142,143]. Myers et al. [42] displayed coumarin’s antiproliferative effects against renal cancer using the 786-O and A-498 RCC cancer lines. Coumarin alone inhibited the growth of these renal cancer cell lines. Combination treatment of coumarin and suramin (a chemotherapeutic agent) further inhibited cell proliferation. However, no data were presented describing the mechanisms behind the antiproliferative effects of coumarin.

In an in vivo analysis, coumarin was tested for antitumor effects in BALB/c mice injected with baby mouse kidney (BMK) cells, which induced tumor formation. Treatment with coumarin (20 mg/kg, three times per week for three weeks) caused a reduction in mean tumor diameter. Additionally, histopathological analysis revealed that no metastases occurred in tumors treated with coumarin [132].

#### 3.1.6. Curcumin

Curcumin (Figure 2) is the primary curcuminoid present in turmeric (*Curcuma longa* L.). Curcumin has gained vast attention worldwide for its wide range of biological activities, such as antioxidant, anti-inflammatory, antimicrobial, antiviral, and anticancer effects [144,145,146,147]. Several studies have investigated the anticancer effects of curcumins in various renal cancer cell lines. In a study performed on RCC Caki cells, curcumin treatment reduced cell viability and induced cytotoxicity. It also resulted in the activation of caspase-3, cleavage of phospholipase C-γ1 and DNA fragmentation. Dephosphorylation of Akt, downregulation of B-cell lymphoma 2 (Bcl-2), B-cell lymphoma-extra-large (Bcl-xL), and inhibitor of apoptosis (IAP) proteins, release of cytochrome c (cyt. c), and caspase-3 activation preceded the induction of apoptosis [43]. In an additional study, Caki cells treated with curcumin and TRAIL resulted in an increased accumulation of sub-G1 phase cells and a ladder pattern of internucleosomal fragmentation. DEVDase activity, death receptor 4 (DR4) and DR5 were all increased post-treatment. Furthermore, pretreatment with N-acetylcysteine (NAC) blocked the curcumin plus TRAIL-induced apoptosis and prevented DNA fragmentation, confirming ROS generation by curcumin. Ectopic expression of peroxiredoxin II inhibited the induction of apoptosis, which blocked the upregulation of DR5. Taken together, the findings showed that curcumin enhanced TRAIL-induced apoptosis by ROS-mediated DR5 upregulation in Caki cells [44]. In a subsequent study, the same investigators treated Caki, ACHN, and A-498 cell lines with curcumin. Although curcumin did not display antiproliferative properties alone, in combination with dactolisib, a chemotherapeutic agent, it was shown to induce apoptosis in a caspase-dependent manner. Furthermore, combined treatment downregulated myeloid cell leukemia-1 (Mcl-1) and Bcl-2 expression [45]. Using the RCC-949 cell line, Zhang et al. [46] exhibited curcumin’s antiproliferative properties. Curcumin was shown to induce cell apoptosis possibly through modulation of expression of Bcl-2 and Bax. Additionally, treatment with curcumin induced G2/M phase arrest and negatively modulated the phosphoinositide 3-kinase/protein kinase B (PI3K/Akt) pathway. Finally, Xu et al. [47] investigated curcumin’s synergistic effect with the chemotherapeutic drug temsirolimus, using Caki-1 and OS-RC-2 cell lines. This combination led to markedly induced apoptosis in RCC cells through an increase in Yes-associated protein (YAP)/p53 expression. Treatment with curcumin alone also increased YAP expression and induced apoptosis, displaying the antiproliferative effects of curcumin in RCC cell lines.

#### 3.1.7. Cyanidin

Cyanidin (Figure 2), an anthocyanin, is a pigment found in various fruits and vegetables, including grapes, blackberries, blueberries, apples, and red onions. It has previously displayed antioncogenic, antiproliferative, and apoptotic properties [148,149]. Liu et at. [48] examined cyanidin’s effect on RCC cell lines 786-O and ACHN, where it was found to inhibit growth and migration via G1/M cell cycle arrest and the induction of apoptosis. Cyanidin increased the expression of early growth response protein 1 (EGR1), while decreasing the expression of selenoprotein W (SEPW1).

Liu et al. [48] additionally investigated cyanidin’s effects in vivo using male BALB/c mice xenografted with ACHN cells. Cyanidin reduced tumor growth, volume, weight, and the expression of intratumor Ki-67. These data corroborated with the in vitro results presented earlier, further displaying cyanidin’s ability to suppress the migration and proliferation of renal cancer cells.

#### 3.1.8. Daphnetin

Daphnetin (Figure 2) is a coumarin derivative extracted from the Daphne Korean Nakai (*Changbai daphne*) plant, and Euphorbia semen, the dried and ripe seed of *Euphorbia lathyris* L. It has previously exhibited anti-inflammatory and anticancer properties [150]. Finn et al. [49] explored daphnetin’s anticancer effects using RCC cell line A-498. Daphnetin exhibited antiproliferative properties and inhibited cell cycle progression, specifically at the S phase. After treatment with daphnetin, p38 was shown to be activated, and expressions of cytokeratin 8 and cytokeratin 18 were increased. Higher concentrations of daphnetin were required to inhibit extracellular signal-regulated kinase 1 (ERK1) and ERK2.

#### 3.1.9. Dicoumarol

Dicoumarol (Figure 2), a symmetrical biscoumarin, is a naturally occurring anticoagulant that was initially isolated from sweet clover hay (*Melilotus officinalis*). Although it is mainly known for its anticoagulant properties [151], dicoumarol has also exhibited anticancer effects [152]. Park et al. [50] found that dicoumarol decreased cell viability in the ACHN, A-498, and Caki RCC cell lines and downregulated Bcl-2 (via inhibition of NF-κB and cyclic-AMP response element-binding protein), Mcl-1, and c-FLIP expression in Caki cells. Furthermore, combined treatment with dicoumarol and TRAIL induced apoptosis in ACHN and A-498 cells.

#### 3.1.10. Esculetin

Esculetin (6,7-dihydroxycoumarin, Figure 2), a derivative of coumarin, is the active ingredient in the Chinese herbal medicine known as Ash Bark (*Cortex Fraxini*) and has shown a potential therapeutic role in various diseases, such as obesity, diabetes, cardiovascular ailments, renal failure, and neurological and neoplastic disorders [153]. Using the 786-O and SN12-PM6 RCC cell lines, Duan et al. [51] observed a decrease in cell viability after treatment with esculetin in a concentration- and time-dependent manner. Esculetin arrested cell cycle progression in the G0/G1 and G2/M phases, induced apoptosis, and inhibited cell migration and invasion. Additionally, cyclin D1, cyclin-dependent kinase 4 (CDK4), CDK6, and c-Myc expressions were decreased, while epithelial cadherin (E-cadherin) expression was increased.

#### 3.1.11. Eupafolin

Eupafolin (6-methoxy-5,7,30,40-tetrahydroxyflavone, Figure 2), a flavone found in Japanese mugwort (*Artemisia princeps*), has been shown to inhibit the growth of multiple human cancer cell lines [154,155,156]. In an in vitro analysis using the Caki RCC cell line, eupafolin alone did not induce apoptosis. However, a combined treatment with eupafolin and TRAIL induced apoptosis via downregulation of Mcl-1 and upregulation of Bcl-2-like protein 11 (Bim) (via AMP-activated protein kinase (AMPK)-mediated inhibition of proteasome activity) [52].

Eupafolin was additionally tested in vivo using male BALB/c mice xenografted with Caki cells. Treatment with eupafolin (10 mg/kg) and TRAIL (3 mg/kg) combined for 21 days suppressed tumor growth and induced cell death. The combination treatment of eupafolin and TRAIL inhibited tumor growth to a greater extent than treatment with eupafolin alone [52].

#### 3.1.12. Gambogic Acid

Gambogic acid (Figure 2), a natural phenolic compound formed from the xanthone backbone, is found in gamboge trees (*Garcinia hanburyi*). It has exhibited various anti-inflammatory, antioxidant, and anticancer effects both in vitro and in vivo [157,158]. Jiang et al. [53] examined the effects of gambogic acid alone and in combination with the chemotherapeutic agent sunitinib in Caki-1 and 786-O RCC cell lines. A combination of gambogic acid and sunitinib exhibited greater antiproliferative activity compared to gambogic acid alone. Furthermore, combination treatment with sunitinib led to a greater increase in the sub-G1 population when compared to monotherapy. Cotreatment led to upregulation of p21, also known as cyclin-dependent kinase inhibitory protein-1, cyclin-dependent kinase inhibitor 1 or CDK-interacting protein 1, inhibition of Bcl-2, and downregulation of VEGF.

Gambogic acid (5 mg/kg) was also tested in vivo using male BALB/c mice xenografted with Caki-1 (clear cell RCC) cells. Tumor-bearing mice were treated with gambogic acid alone (5 mg/kg), sunitinib alone (20 mg/kg), and combination treatment of gambogic acid and sunitinib. Tumor volume and weight were decreased substantially more by combination treatment of gambogic acid and sunitinib than with gambogic acid alone. While gambogic acid alone decreased Ki-67, cluster of differentiation 31 (CD31), and nuclear p65 expression (involved in the NF-κB signaling pathway), cotreatment decreased these proteins to a greater extent [53].

#### 3.1.13. Genistein

Genistein (Figure 2), a polyphenolic isoflavone, can be found in soybeans, fava beans, and lupin and various other plants, such as Sohphlang (*Flemingia vestita*) and large leaf flemingia (*Flemingia macrophylla*). It has previously shown encouraging anticancer results in various human cancer cell lines and animal tumor models [159,160]. In one of the earlier studies, genistein was tested against various renal cancer cell lines, such as SMKT-R1, SMKT-R2, SMKT-R3 and SMKT-R4. Treatment with genistein inhibited proliferation and induced apoptosis in the RCC cell lines in a concentration- and time-dependent manner. Specific mechanisms were not established for these anticancer effects [54]. Majid et al. [55] found that genistein treatment had antiproliferative effects in cell lines A-498 and ACHN. B cell translocation gene 3 (BTG3) was shown to be epigenetically silenced in RCC, while treatment with genistein reactivated BTG3. Genistein induced BTG3 expression through promoter demethylation and active histone modification. Additionally, treatment with genistein led to G2/M cell cycle arrest, downregulation of DNA methyltransferase (DNMTase) activity, and decreased histone deacetylase activity. Hirata et al. [56] investigated the effects of genistein on RCC cell lines A-498, 786-O, and Caki-2. Genistein was found to inhibit proliferation and invasion and to promote apoptosis. miR-1260b was downregulated, suggesting that genistein inhibited Wnt signaling. Genistein also reduced TOPflash reporter activity in RCC cells and decreased luciferase activity in secreted frizzled related protein 1 (sFRP1), Dickkopf WNT signaling pathway inhibitor 2 (Dkk2), and SMAD Family Member 4 (Smad4) target genes.

Sasamura et al. [54] tested genistein’s antiangiogenic effect in vivo using a female mouse (dorsal air sac model). A portion of SMKT R-1 (RCC) cells were pretreated with genistein (100 μg/mL) for 12 h, while another portion of SMKT R-1 cells were not subjected to pretreatment with genistein before they were injected into the Millipore filter chamber. The median vascular volume with and without genistein pretreatment decreased by 56.4 and 48.9%, respectively. Although the mechanisms were not confirmed, the decrease in vascular volume was postulated to be caused by inhibition of mRNA expression of VEGF and basic fibroblast growth factor.

#### 3.1.14. Hispidulin

Hispidulin (4′,5,7-trihydroxy-6-methoxyflavone, Figure 2) is a naturally occurring flavonoid present in several Chinese medicinal herbs, particularly *Saussurea involucrate*, commonly referred to as “snow lotus”. Hispidulin has been found to exhibit anticancer activities against various cancer cells and models [161,162]. Gao et al. [57] observed that when used alone against 786-O and Caki-1 RCC cell lines, hispidulin inhibited proliferation by increasing G0-G1 phase arrest, decreasing G2 phase, increasing caspase-3 activity, and downregulating Bcl-2, survivin and phosphorylated signal transducer and activator of transcription 3 (p-STAT3). When used in combination alongside sunitinib, a VEGF-R2 and VEGF-R3 receptor tyrosine kinase inhibitor commonly used in anticancer treatment of gastrointestinal stromal tumors, hispidulin required relatively low concentrations to exhibit antitumor activity compared to its use individually. Subsequently, the same research team [58] further examined the mechanisms of hispidulin in exhibiting growth inhibition, apoptosis induction and cell cycle arrest in human clear cell RCC. Hispidulin treatment of Caki-2 and ACHN RCC cells induced apoptosis as observed through decreased sphingosine kinase 1 (SphK1) and increased ceramide levels. These results indicate that hispidulin could be an effective treatment against renal cancer.

Hispidulin was additionally tested both alone and in combination with sunitinib in vivo using male BALB/c mice xenografted with Caki-1 cells. The tumor-bearing animals treated with hispidulin, both alone and in combination with sunitinib, showed significantly decreased tumor growth and size compared to the control. Treatment with hispidulin (20 mg/kg) in combination with sunitinib showed decreased Ki-67 labeling index and decreased microvessel density compared to RCC cells treated with sunitinib alone, confirming hispidulin’s efficacy as an antitumor treatment [57]. Gao et al. [58] subsequently tested hispidulin in vivo using male BALB/c mice xenografted with Caki-2 cells. Treatment with hispidulin (20 mg/kg) exhibited increased cell apoptosis via increased caspase-3 and phosphorylated c-Jun NH_2_-terminal kinase (p-JNK) expression, decreased SphK1 activity, and ceremide accumulation intracellularly, confirming hispidulin’s efficacy as an antitumor treatment.

#### 3.1.15. Honokiol

Honokiol (Figure 2) is a dormant biphenolic compound found naturally in the leaves and bark of Houpu magnolia (*Magnolia officinalis* Rehder & E. Wilson). This multi-targeted phytochemical has shown significant promise for prevention and treatment of various cancers [163,164]. Li and colleagues [59] evaluated honokiol’s efficacy against RCC cancer cell line A-498. It was found that honokiol suppressed proliferation and migration of the A-498 cell line by dual-blocking the epithelial–mesenchyme transition and reversing the miR-141/zing finger E-boc binding homebox 2 (ZEB2) axis, a tumor suppressor that is downregulated in RCC. In a separate study, honokiol was found to effectively decrease cell proliferation, migration, and invasion in the extremely metastatic 786-O RCC cell line. Honokiol treatment resulted in increased levels of activated RhoA GTPase, further activating and resulting in increased levels of downstream Rho-associated protein kinase (ROCK). Subsequently, ROCK directly phosphorylated myosin light chain, leading to inhibition of MLC phosphatase with a reduction in migration of RCC cells [60].

Honokiol was additionally tested in vivo using BALB/c mice xenografted with A-498 cells. The tumor-bearing animals treated with honokiol showed significantly decreased tumor growth and size compared to the control. Additionally, honokiol-treated animals exhibited decreased ZEB2, vimentin and fibronectin levels and increased E-cadherin levels in tumor tissues compared to the control [59]. It can be concluded that honokiol has the potential for use as an effective anticancer therapy for renal cancer.

#### 3.1.16. Hymecromone

Hymecromone (4-methylumbelliferone, Figure 2), a widely studied coumarin found in *Dalbergia volubilis* (commonly known as climbing Dalbergia) and *Galactia regularis* (commonly known as milk pea), is a hyaluronic acid inhibitor and a dietary supplement sold in Europe and Asia [165]. Hymecromone has shown potential for the treatment of inflammatory disorders, immune diseases, and cancer [166,167]. Benitez and colleagues [61] found that when used alone, hymecromone was ineffective in reducing metastasis of Caki-1, ACHN, 786-O, and A-498 RCC cell lines. However, when used in combination with sorafenib, an angiogenesis inhibitor used for RCC treatment, it showed synergistic effects and potent anticancer properties against the RCC cell lines compared to sorafenib alone, which showed no effectiveness. Hymecromone and sorafenib in combination downregulated cluster of differentiation 44 (CD44), receptor for hyaluronan-mediated motility (RHAMM), p-MEK, p-ERK, and p-VEGFR levels. These results support the potential use of hymecromone as a supplementing anticancer therapy to sorafenib to increase its efficacy and the long-term survival of RCC patients.

#### 3.1.17. Icaritin

Icaritin (Figure 2), a naturally occurring flavonoid derivative present in *Epimedium brevicornum*, commonly known as horny goat weed, is used in traditional Chinese medicine [168]. Icaritin exhibits anticancer properties through antiproliferative and proapoptotic mechanisms, including suppression of osteoclast differentiation, apoptosis induction, and ERK signaling inhibition [168]. Li et al. [62] found that icaritin reduced metastasis in RCC 786-O cell lines via its ability to suppress activated STAT3, an independent prognostic indicator of RCC. Icaritin showed effectiveness in combating tumorigenesis across multiple pathways, including inhibiting constitutive and IL-6 induced STAT3 and reducing levels of antiapoptotic and pro-proliferative proteins Bcl-xL, Mcl-1, survivin, cyclin E and cyclin D1. Icaritin exhibited STAT3 suppression and subsequent apoptosis induction through its ability to inhibit activation of upstream Januse kinase 2 (JAK2) levels, which prevents phosphorylation and consequent activation of STAT3. Icaritin’s JAK2 suppressive properties also provide a potential mechanism of its ability to modestly inhibit IL-6-induced Akt and mitogen activated protein kinase (MAPK) pathways. Additionally, icaritin increased the levels of cleaved caspase-3 and cleaved PARP, demonstrating its pro-apoptotic effect.

Icaritin was additionally tested in vivo using female BALB/c mice xenografted with Renca cells. Treatment with icaritin (10 mg/kg) once every other day was found to exhibit antitumor effects mediated by reduced STAT3 activity as well as Bcl-xL and cyclin E expression. Furthermore, icaritin treatment reduced VEGF expression, indicating its potential for use as an antiangiogenic agent [62]. These results indicate that icaritin holds great potential for combating RCC.

#### 3.1.18. Isoliquiritigenin

Isoliquiritigenin (2′,4′,4-trihydroxychalcone, Figure 2), a chalcone flavonoid naturally found within licorice roots and shallots, has been found to have antioxidant, anti-inflammatory and antitumor activities [169]. Using isoliquiritigenin-treated Caki RCC cell lines, Kim et al. [63] observed decreased cell viability through an increased ROS production. p-STAT3 was found in high levels in RCC; however, isoliquiritigenin-treated RCC showed markedly diminished phosphorylated STAT3 levels, decreasing its binding activity, leading to decreased production of cyclin D1 and cyclin D2 levels and decreased cleavage of caspase-3, caspase-7, caspase-9, and PARP. Isoliquiritigenin decreased upstream phosphorylation and activation of JAK2, further preventing activation of STAT3. Isoliquiritigenin also exhibited proapoptotic activity through downregulation of protooncogene murine double minute 2 (Mdm2), which induced increased p53 activity. Additionally, Bax and cyt. c release expressions were increased, and expression of Bcl-2 and Bcl-xL was decreased after isoliquiritigenin treatment, further enforcing its suppressive effects against renal carcinogenesis.

#### 3.1.19. Luteolin

Luteolin (3′,4′,5′,7′-tetrahydroxyflavone, Figure 2), a natural flavone flavonoid, is found in many fruits and vegetables, including celery, sweet bell peppers, carrots, onion, broccoli, parsley and chamomile tea, and it is commonly used against inflammatory diseases [170,171]. Ou et al. [64] found that when used alone in 786-O, A-498 and ACHN RCC cell lines, luteolin exhibited cytotoxicity and apoptotic activity through downregulation of Akt and activation of p38, JNK, and ERK. Through decreased p-Akt levels, it was also proposed that luteolin decreased levels of heat shock protein 90 (HSP90), an ATPase-directed chaperone protein demonstrated in RCC. In a follow-up study by the same group [65], 786-O, A-498, and ACHN RCC cells were treated with luteolin in combination with TRAIL. The luteolin- and TRAIL-treated cells showed increased apoptosis compared to TRAIL used alone through increased levels of caspase-3, caspase-8, and caspase-9. Luteolin potentiated TRAIL-induced cytotoxicity, and the combination exhibited markedly reduced Mcl-1, FLIP, STAT3, and Akt levels in the RCC cells when compared to TRAIL alone. Evidence of luteolin’s effectiveness both alone and in combination indicates its antiproliferative effects in RCC cell lines.

#### 3.1.20. Osthole

Osthole (7-methoxy-8-(3-methyl-2-butenyl) coumarin, Figure 2) is a naturally occurring monomeric coumarin found in high concentrations in the *Cnidium monniere* plant, also known as snow parsley, and other Chinese medicinal plants, such as *Angelica pubescens*, *Archangelica*, *Citrus*, and *Clausena* [172]. It has been found to have multiple pharmacological properties, including antioxidant, anti-inflammatory, immunomodulatory, and anticancer activities [173,174,175]. Liu et al. [66] found that when used alone in 786-O and ACHN RCC cell lines, osthole suppressed proliferation and colony formation through increased expression of proapoptotic caspase-3 and Bax protein levels and decreased expression of antiapoptotic Bcl-2 and surviving levels. This study also demonstrated osthole’s ability to reduce migration through decreased expression of invasion-related protein matrix metalloproteinase (MMP) 2 and MMP 9. In a subsequent study by Min et al. [67], osthole was used in combination with TRAIL to explore its effect against RCC. The combination treatment of Caki RCC cells showed markedly increased apoptosis compared to TRAIL alone. This was exhibited through downregulation of c-FLIP and increased cyt c release. Evidence of osthole’s effectiveness both alone and in combination indicates its antitumor and antimetastatic effects in RCC and its effectiveness as a TRAIL sensitizer.

#### 3.1.21. Praeruptorin B

Praeruptorin B (Figure 2), a naturally derived compound found in *Peucedanum praeruptorum*, *Saposhnikovia divaricata*, and *Angelica cincta* plants commonly used in Chinese medicine, is known to exhibit potent antitumor properties [176]. Praeruptorin B displayed cytotoxic effects against 786-O and ACHN RCC cell lines as observed by Lin et al. [177]. Praeruptorin B was found to potently inhibit migration and invasion of the RCC cells, as evidenced by decreased production of cathepsin C (CTSC) and cathepsin V (CTSV) proteins, proteases that regulate cancer cell invasion extracellularly. Praeruptorin B treatment of RCC cells also exhibited reduced levels of p-EGFR, phosphorylated mitogen activated protein kinase (p-MEK), and phosphorylated extracellular signal-related kinase (p-ERK), and suppression of the EGFR-MEK-ERK signaling pathway. Evidence of praeruptorin B’s ability to reduce migration and invasion effectively indicates its antimetastatic capabilities.

#### 3.1.22. Pterostilbene

Pterostilbene (trans-3,5-dimethoxy-4-hydroxystilbene, Figure 2) is a naturally derived compound found primarily in blueberries, but also in grapes and heartwood (*Pterocarpus marsupium*), that is commonly used for its antioxidant and anti-inflammatory properties [178,179,180]. Zhao et al. [69] found that treatment with pterostilbene in A-498 and ACHN RCC cell lines potently induced cytotoxicity and apoptosis through upregulation of proapoptotic markers cyt. c, Bad, Bak, Bax, caspase-3, caspase-9, and PARP and downregulation of antiapoptotic Bcl-2. Additionally, pterostilbene induced S-phase cell cycle arrest and downregulated p-Akt and p-ERK1 and p-ERK2. Pterostilbene-treated RCC cells had increased expression of γH2AX protein levels, indicating an increase in double-stranded DNA breaks, and decreased expression of Rad51, exhibiting inhibition of homologous recombination.

#### 3.1.23. Resveratrol

Resveratrol (*trans*-3, 5, 4′-trihydroxystilbene, Figure 2) is the primary polyphenolic compound found in grape skin, blueberries, and peanuts. Resveratrol is well-known for its antioxidant, antifungal, and anti-inflammatory properties and has been reported to be effective in combating human malignancies [181,182,183]. Several studies have investigated the anticancer effects of resveratrol in various renal cancer cell lines. In a study performed on RCC RCC5430 cells, resveratrol inhibited cell growth and induced cell death. These effects were mediated by upregulation of vitamin D receptor (VDR) and tumor necrosis factor receptor associated factor 1 (TRAF-1), downregulation of cell surface receptor glial family receptor alpha 2 (GFRA2) and hemopoietic cell protein-tyrosine kinase (HCK), and repression of the G-protein coupled receptor chemokine receptor 4 (CXCR4), compared to control RCC cells [70]. In another study, 786-O RCC cells treated with resveratrol showed reduced proliferation via decreased VEGF expression [71]. Resveratrol further demonstrated impaired cell viability, migration, and invasion in ACHN RCC cells via increased histone acetylation through degradation of proteolytic enzymes MMP-2 and MMP-9, considered key proteins in tumor cells’ ability to disrupt cell-to-cell interaction and remodel tissue [72]. Liu et al. [73] observed that Ketr-3 RCC cell viability and migration, in addition to apoptosis, were affected by resveratrol treatment. This study also documented a decreased expression of antiapoptotic Bcl-2 protein and autophagy related (ATG5 and ATG7) genes, increased expression of p53 and AMPK, and decreased phosphorylation of mTOR. In a separate study conducted by Zhao et al. [74], ACHN and A-498 RCC cells treated with resveratrol were found to have decreased invasion and cell migration via decreased expression of vimentin, Snail, MMP-2, MMP-9, p-Akt, and p-ERK1/2 and increased expression of E-cadherin and tissue inhibitor of metalloproteinase 1 (TIMP-1). A subsequent study showed inhibition of induction of apoptosis in resveratrol-treated ACHN and 786-O RCC cell lines as observed through significantly downregulated expressions of NLR family pyrin domain containing 3 (NLRP3), an inflammasome, leading to decreased anti-apoptotic caspase-1 expression and increased pro-apoptotic caspase-3 expression [75]. Resveratrol was also evaluated for its anticancer properties on Caki-1 RCC cells treated with increasing concentration of paclitaxel, an anticancer drug that hyperstabilizes polymerized microtubules. The cells treated with resveratrol were found to have decreased survivin levels, which further increased the sensitivity of paclitaxel-resistant RCC cells to paclitaxel treatment [76]. Finally, resveratrol was evaluated for its effects in combination with sorafenib on Caki-1 and 786-O RCC cell lines. Kim et al. [77] found that resveratrol showed synergistic activity with sorafenib by blocking STAT3 and STAT5 activation, thus inducing S phase arrest and apoptosis via loss of mitochondrial membrane potential.

Resveratrol was additionally tested in vivo using female BALB/c mice xenografted with Renca cells. The tumor-bearing animals treated with resveratrol (2.5 and 5 mg/kg) showed significantly decreased tumor growth and size compared to the control. Additionally, resveratrol treated animals exhibited increased FasL, perforin, and granzyme B levels in tumor tissues compared to the control [133]. Resveratrol was subsequently tested in vivo using male nude mice xenografted with ACHN cells. Treatment with resveratrol (60 mg/kg) was found to induce apoptosis via increased Bax:Bcl-2 expression ratio and decreased NLRP3 expression [75]. Moreover, resveratrol was tested in vivo alone and in combination with sitagliptin, a DDP-4 enzyme inhibitor used in treatment of diabetes mellitus, using male Wistar mice with clear cell RCC induced by a single intraperitoneal administration of DEN (200 mg/kg). The tumor-bearing animals treated with resveratrol showed increased apoptosis via STAT3 signaling and nuclear factor-like 2 (Nrf2)/heme oxygenase-1 (HO-1) pathway reduction. These effects were significant when sitagliptin and resveratrol were used in combination compared to use of each of these drugs alone [134]. It can be concluded that resveratrol has the potential for use as an effective therapy for renal cancer.

#### 3.1.24. Umbelliferone

Umbelliferone (7-hydroxycoumarin, Figure 2), a polyphenolic compound found extensively in the Rutaceae and Apiaceae family herbs and fruits, including golden apples, carrots, sanicle, angelica, celery, cumin, fennel, and parsley, is known for its antibacterial, anti-inflammatory, and anticancer activities [184,185]. Wang et al. [78] observed that when used against ACHN, 786-O, and OS-RC-2 RCC cell lines, umbelliferone inhibited cell proliferation by inducing G1 cell cycle arrest, increasing Bax protein and decreasing Ki67, MCM2, Bcl-2, cyclin-dependent kinase 2 (CDK2), cyclin E1, CDK4, and cyclin D1 protein levels. Umbelliferone demonstrated inhibition of p110γ, an isoform of catalytic subunit of PI3K, which contributes to its ability to regulate the cell cycle and induce apoptosis. These encouraging results underscore umbelliferone’s effectiveness as a potential therapeutic agent for RCC treatment.

### 3.2. Terpenes and Terpenoids

#### 3.2.1. Anatolicin

Tosun et al. [79] isolated anatolicin (Figure 3), a sesquiterpene coumarin ether, from the root extract of *Heptatera anatolica* for the first time. Anatolicin was tested against UO31 kidney cancer cell lines and showed high cytotoxic effects with unknown mechanisms [79].

#### 3.2.2. Artemisinin

Artemisinin (Figure 3), a terpenoid derived from *Artemisia annua* L., has been found to possess anti-inflammatory, immunomodulatory and anticancer properties [186,187]. Yu et al. [80] investigated the effects of artemisinin on clear cell RCC cells to elucidate the potential of this terpenoid as an anticancer agent. The results of the study found that artemisinin inhibited tumor cell proliferation, migration, and invasion and suppressed the Akt and ERK1/2 signaling pathways. Additionally, clear cell renal carcinoma cells were treated with a combination of artemisinin and Akt inhibitor VIII (a pan-Akt inhibitor) to enhance the anticancer effects of artemisinin. It was found that a combined treatment suppressed clear RCC cell colony formation in UMR-C and Caki-2 cell lines. Based on RT-qPCR analysis, artemisinin was also found to decrease mRNA levels of c-Myc, cyclin D1, PCNA, N-cadherin, Vimentin, and Snail, and increased mRNA levels of E-cadherin in both cancer cell lines. Artemisinin also inhibited clear cell RCC cell migration and invasion by reducing the ability of the UMRC-2 and CAKI-2 cells to pass through the basement membrane.

Yu et al. [80] also studied the in vivo effects of artemisinin using a UMRC-2 xenograft tumor model. UMRC-2 cells were subcutaneously injected into nude mice, which were divided into a control group and treatment group. At post-injection day 7, the mice were treated with 20 mg/kg every day for 14 days. Results found that the tumor size and weight in the artemisinin treatment group were lower and exhibited a slower rate of tumor growth compared to the control. Mechanistic studies using tumor tissues showed that artemisinin decreased mRNA levels of c-Myc, cyclin D1, PCNA, N-cadherin, Vimentin and Snail, but increased mRNA levels of E-cadherin. Additionally, tumors treated with artemisinin suppressed the phosphorylation of Akt compared to control animals.

#### 3.2.3. Artesunate

Artesunate (Figure 3) is an anti-malarial drug that originates from artemisinin, which is a sesquiterpene trioxane lactone peroxide derived from the Chinese herb *Artemisia annua*, also called sweet wormwood [188,189]. Artesunate is known to have anticancer activity that has modulatory effects on cancer cell proliferation, invasion, and migration [188,189]. Chauhan et al. [81] studied the efficacy of artesunate against three kidney cancer cell lines originated from papillary cell carcinoma and RCC. The results suggested that artesunate induced cell death and generation of ROS in a receptor-interacting protein kinase 1 (RIP-1)-dependent manner.

#### 3.2.4. Auraptene

Auraptene (Figure 3), a monoterpene coumarin, is commonly found in Seville orange (*Citrus aurantium*) and Bael fruit (*Aegle marmelos*) [190,191]. Auraptene is known to exhibit anticancer effects by modulating various signaling pathways that can affect proliferation, growth factors, and apoptosis [190,191]. Jang et al. [82] examined the suppressive effects of auraptene on cancer progression using RCC4 renal cancer cells. Various key enzymes in the glycolytic pathway, such as glucose transporter 1 (GLUT1), lactate dehydrogenase (LDH), hexokinase 2, and phosphofructokinase, were reduced in renal carcinoma cells treated with auraptene. The study further exhibited the ability of auraptene to inhibit angiogenesis by inhibiting vascularization through VEGF, as well as impair motility in the RCC4 cell line.

Jang et al. [82] studied the in vivo effects of auraptene using an RCC4 xenograft model. Using a Matrigel plug assay, a mixture of Matrigel, VEGF, and either dimethyl sulfoxide (DMSO) or 100 μM of auraptene was subcutaneously injected into nude mice. At 7 days post-injection, the plugs were detached from the skin and visualized. The plugs with auraptene exhibited decreased infiltration of blood vessels with a 5-fold lower hemoglobin content and lower density of infiltrated blood vessels. Using qPCR analysis, the study found that auraptene led to reduction in angiogenesis, displayed by decreased *Vegf-a* mRNA levels. Additionally, the study investigated whether auraptene inhibited angiogenesis by subcutaneously injecting 1 × 10^7^ RCC4 into nude mice. At 9 days post-injection, one group was treated with auraptene every other day. Results showed that auraptene had reduced tumor growth and vascularized vessels. Additionally, immunohistochemical analysis showed that auraptene reduced the number of CD31-positive cells in tumor tissue.

#### 3.2.5. β-Elemene

β-elemene (Figure 3), a sesquiterpene extracted from the herbs *Curcuma Rhizoma* and *Curcuma wenyujin*, has been used in traditional Chinese medicine for thousands of years [192,193]. β-elemene has been shown to have anticancer potential by inhibiting cell proliferation, inducing cell cycle arrest, and promoting apoptosis [192,193]. Zhan et al. [83] showed that β-elemene inhibited the viability of 786-O cells by inducing apoptosis, with downregulation of Bcl-2 and survivin levels. The study found that β-elemene inhibited MAPK/ERK and PI3K/Akt/mTOR signaling pathways.

#### 3.2.6. Betulin

Betulin (Figure 3), a lupane pentacyclic triterpene compound, is derived from many different plants, such as the bark of birch trees, specifically the Betulaceae family [194]. It can also be extracted from various parts of plants belonging to the Platanaceae, Dilleniaceae, Rhamnaceae, Rosaceae, and Fagaceae families [194]. Betulin has been shown to have anti-inflammatory and anticancer effects such as the induction of mitochondrial oxidative stress, regulation of specific protein transcription factors, and the inhibition of signal transducers [194,195,196]. Cheng et al. [84] evaluated betulin’s effect on RCC 786-O and Caki-2 cell lines, where it was found to inhibit cell proliferation and decrease cell viability. Treatment with betulin dramatically reduced the number of colonies of 786-O cells. Betulin was additionally examined for its interaction with the mTOR signaling pathway. mTOR inhibitors are known to block the dysregulation of tuberous sclerosis 1 (TSC1) or tuberous sclerosis 2 (TSC2) in RCC cells and are a promising treatment for RCC. The RCC cell line 786-O was more sensitive to betulin in comparison to mTOR-inactive Caki-2 cells. Betulin increased mTOR activation, which in turn, decreased cell viability in the 786-O cell lines. Additionally, betulin decreased aerobic glycolysis, decreased the expression of phosphorylated ribosomal protein S6 (p-S6), p-EBP1, pyruvate kinase muscle isozyme M2 (PKM2), and HK2 proteins in the 786-O cell line, but not Caki-2 cell line. These findings exhibit the therapeutic potential of betulin treatment in RCC patients with mTOR activation.

#### 3.2.7. Betulinic Acid

Betulinic acid (Figure 3), a pentacyclic triterpene isolated from various plants, such as birch bark *Eucommia ulmoides* and white mulberry bark, has been shown to exhibit cytotoxic effects, inhibit proliferation, elicit apoptosis, and repress migration of various cancer cells [195,197]. Yang et al. [85] studied the effects of betulinic acid on RCC and found it to inhibit RCC proliferation and induce apoptosis via upregulation of Bcl-2-associated X protein, cleaved caspase-3, ROS, MMP-2, MMP-9, and vimentin and downregulation of Bcl-2, MMP-2, and E-cadherin in renal cancer cells.

Yang et al. [85] also studied the in vivo effects of betulinic acid. Nude mice were subcutaneously injected with 786-O and ACHN cell lines and then administered betulinic acid at doses of 5 and 10 mg/kg for 15 days. Both doses of 5 mg/kg and 10 mg/kg betulinic acid significantly slowed growth of tumors, which was measured by volume compared to the control group at day 15 post treatment. Immunohistochemistry staining was also performed on tumor samples, and the results showed that there were fewer Ki-67- and MMP-9-positive cells in tumors that were treated with betulinic acid compared to the control.

#### 3.2.8. Cafestol

Cafestol (Figure 3), a diterpene compound found in the beans of coffee (*Coffea arabica*), has been reported to possess various pharmacological activities, including anti-inflammatory, anticarcinogenic and antitumor effects [198]. Choi et al. [86] studied the anticancer effect of cafestol in Caki RCC cells. Cafestol was found to inhibit the viability of Caki cells by inducing apoptosis via an increase in proapoptotic proteins, caspase-2 and caspase-3, Bim, and Bax, and a decrease in antiapoptotic proteins, such as cFLIP, Bcl-2, Mcl-1 and Bcl-xL. Additionally, the results showed that cafestol inhibited the PI3K/Akt signaling pathway. Overall, this study showed that cafestol modulates apoptosis mechanisms in human renal Caki cells. ABT-737 is a cancer therapeutic agent that induces apoptosis by inhibiting Bcl-2/Bcl-xL. The therapeutic effects of ABT-737 are blocked by cancer cells that express high levels of Mcl-1 [199]. To enhance ABT-737-mediated apoptosis, Woo et al. [87] examined whether cafestol could stimulate ABT-737-mediated apoptosis in Mcl-1-overexpressed Caki cells. Since Mcl-1-overexpressed Caki cells are resistant to ABT-737, there was no effect of ABT-737 solely on apoptosis. However, when cafestol was used in combination with ABT-737, there was significantly increased apoptosis in the Caki/Mcl-1 cells. The main mechanism of the combination treatment-induced apoptosis is through the downregulation of Mcl-1 and upregulation of Bim.

Woo et al. [87] also studied the in vivo effects of cafestol or ABT-737 treatment individually, and then in combination. Nude mice were subcutaneously injected with Caki/Mcl-1 cells and then administered cafestol alone, ABT-737 alone, or combined cafestol and ABT-737 for 14 days. Combined treatment significantly suppressed tumor growth by decreasing tumor size at the end of the 14 days, compared to cafestol or ABT-737 treatment alone. Furthermore, terminal deoxynucleotidyl transferase (TdT)-mediated dUTP nick-end labeling showed that combined treatment increased cell death, and immunohistochemical staining of the tumors showed that combined treatment had increased the activated caspase-3 levels.

#### 3.2.9. Carsonic Acid

Carsonic acid (Figure 3), a phenolic diterpene found in rosemary (*Rosmarinus officinalis*) and Holy Basil or tulsi (*Ocimum sanctum*), is known to have anti-inflammatory, antiviral, and antitumor activities [200,201]. Jung et al. [88]. investigated the effect of carsonic acid on TRAIL-mediated cell viability and apoptosis and molecular mechanisms in human renal carcinoma Caki cells. The results showed that carsonic acid induced TRAIL sensitization in cancer cells by sensitizing apoptosis and decreasing cell viability. It was found that carsonic acid in combination with TRAIL treatment induced caspase-3, caspase-8, and caspase-9 activation, sensitizing TRAIL-mediated apoptosis through caspase activation. Additionally, carsonic acid and TRAIL treatment combinations reduced mitochondrial membrane potential levels and increased cytosolic cyt. c level through Bax activation in renal carcinoma cells, which decreased the integrity of the mitochondrial membrane potential. The carsonic acid and TRAIL combination similarly increased the expression of proapoptotic proteins, such as DR5, Bim and p53 upregulated modulator of apoptosis (PUMA), while downregulating antiapoptotic proteins, such as c-FLIP and Bcl-2. Park et al. [89] studied carsonic acid for its anticancer activities on Caki RCC cells. This study found that carsonic acid significantly reduced Caki cell viability by inducing apoptosis in an assay. It was further observed that carsonic acid induced apoptosis through mitochondrial pathways by inducing activation of caspase-9, caspase-7, and caspase-3 and cleavage of PARP. Additionally, carsonic acid was found to downregulate Bcl-2 and Bcl-xL. Finally, carsonic acid was also found to diminish the constitutive phosphorylation of STAT3 signaling.

#### 3.2.10. Corosolic Acid

Corosolic acid (Figure 3), a pentacyclic triterpenoid isolated from Banaba (*Lagerstoemia speciose*) leaves, is known to have anticancer effects via regulation of proliferation, apoptosis, and angiogenesis of tumor cells [202]. Woo et al. [90] performed a study to determine if corosolic acid could induce cell death in Caki renal carcinoma cells. The results showed that corosolic acid decreased cell viability and increased cytotoxicity in Caki renal carcinoma cells. Moreover, corosolic acid-induced cell death was found to be associated with elevated lipid peroxidation and ROS levels. The study also found that corosolic acid induced caspase-independent non-apoptotic cell death that was also not associated with necroptosis.

#### 3.2.11. Crocin

Crocin (Figure 3) is a carotenoid and one of the major constituents derived from saffron, *Crocus sativus* [203]. Crocin has been shown to have anticancer effects such as inhibiting tumor growth, inducing cell death, and inhibiting tumor invasion [204,205]. Niu et al. [91] evaluated the anticancer potential of crocin in renal carcinoma cell lines. Crocin was found to inhibit cell migration and viability of treated A498 cells. It increased the expression of miR-577, which acts as a tumor suppressor. The researchers also found that nuclear factor I B (NFIB), a direct target gene of miR-577, was suppressed by crocin treatment, demonstrating its antiproliferative effect.

#### 3.2.12. Cucurbitacins

Cucurbitacins are a group of tetracyclic triterpenoids produced by the plants of the *Cucurbitacaeae* family, and additionally have been found in other families, such as *Scrophulariaceae*, *Polemoniaceae*, and *Thymlaeaceae* [206,207]. Cucurbitacins are known to have cytotoxic properties and anticancer mechanisms of action, such as suppression of proliferation, inhibition of migration and invasion, induction of cell cycle arrest, and apoptosis [206,207]. Henrich et al. [92] investigated the effects of seven different cucurbitacins (cucurbitacin B, C, D, E, I, K, and P) in enhancing TRAIL-mediated apoptosis on renal carcinoma cells. It was found that six of the seven cucurbitacins (cucurbitacins B, C, D, E, I, and K) sensitized ACHN cells to TRAIL, with cucurbitacin P having no significant effect on ACHN cells. Cucurbitacin B was found to be the most potent of the six effective phytochemicals. It was observed that sensitization of ACHN cells by the six effective cucurbitacins led to enhanced TRAIL-dependent activation of caspase-8. Additionally, cucurbitacin B treatment reduced cell number even with general caspase inhibitor ZVAD-FMK, which indicated that cucurbitacins can work via a non-apoptotic cell death or cytostasis mechanism. Cucurbitacin B was also found to produce dramatic changes in cell shape, which included changes in actin distribution, clumping of cells, and decreased levels of phophopaxillin.

#### 3.2.13. Englerin A

Englerin A (Figure 3), a terpenoid derived from the bark of the southeast African plant *Phyllanthus engerli*, has previously exhibited anticancer effects [208]. Ratnayake et al. [93] investigated the effects of englerin A on several renal cancer cell lines and found that englerin showed a 1000-fold selectivity against the seven renal cancer cell lines 786-O, A498, ACHN, Caki-1, RXF-393, SN12C, and UO-31 compared to the 34 natural product extracts studied. The mechanism of inhibited growth of renal cancer cells was not identified in this study. Sulzmaier et al. [94] investigated the cytotoxic activity and growth inhibition of englerin A against human RCC lines A498 and UO-31. The results showed that englerin A selectively decreased cancer cell viability by inducing necrosis in renal carcinoma cells via an increased production of ROS and calcium influx into the cytoplasm. Based on a study conducted by Williams et al. [95], englerin A was found to induce apoptosis and induce necrosis in renal cancer cell lines. Englerin A induced caspase-independent apoptosis and increased autophagic vesicles in A498 cells. In this study, englerin A was also found to accumulate cells in the G2 phase of the cell cycle and blocked the G2/M transition. Englerin A was also found to inhibit Akt and ERK kinases. Carson et al. [96] studied englerin A for its sensitivity to cancer cell growth inhibition against various cell lines, and the study showed that englerin A sensitivity was greatest in cell lines that had transient receptor potential cation channel 4 (TRPC4) expression. These findings confirm that englerin is a TRPC4 agonist, which suppresses tumor cell proliferation, but the mechanism by which TRPC4 channel agonism leads to growth inhibition in tumor cell lines is not known. Batova et al. [97] investigated inflammatory response and tumor growth inhibition of englerin A against clear cell RCC cell lines. The results of the study showed that englerin A altered lipid metabolism by increasing toxic ceramides. Englerin induced endoplasmic reticulum (ER) stress signaling and disrupted the morphology of the ER. This also created an anti-inflammatory response within the cells, mediating antitumor activity.

#### 3.2.14. Escin

Escin (also known as β-aescin, Figure 3), a triterpenoid saponin derived from the horse chestnut tree (*Aesculus hippocastanum*), has been shown to possess antineoplastic activities [209]. Yuan et al. [98] evaluated the cytotoxic impact of escin on renal cancer cell lines 786-O and Caki-1. The results found that escin caused cell death in human renal cancer cells through the mitochondrial apoptosis pathway, which induced G2/M cell cycle arrest and ROS. Escin induced G2/M arrest, and caspase-dependent apoptosis in 786-O cells was activated by increased levels of caspase-9, caspase-8, caspase-3, and caspase-7.

#### 3.2.15. Helenalin

Helenalin (Figure 3), a sesquiterpene lactone found in the flowers of wolf’s bane (*Arnica montana*) and Chamisso arnica (*Arnica chamissonis*), has been shown to have anti-inflammatory, immunomodulatory, and anticancer activities [210,211]. Although it has been used for many years as a treatment for minor injuries in folk medicine, its ability to cause allergic reactions poses a challenge for its use as an anticancer drug [212]. Helenalin was investigated for its anticancer effects on human RCC Caki cells. Exposure of cells to helenalin reduced cell viability via induced accumulation of a sub-G1 population, cleavage of PARP, and apoptotic cell death. Additionally, treatment with helenalin induced apoptosis through increased caspase-3 activation and intracellular ROS levels. Although helenalin induced cellular ER stress through an increase in activation of transcription factor-4 (ATF4), regulation in development and DNA damage responses (REDD1), and CCSST enhancer-binding protein-homologous protein 1 (CHOP) levels, apoptosis is independent of helenalin-induced ER stress [99].

#### 3.2.16. 16-Hydroxyclerod-3,13-dien-15,16-olide (CD)

16-hydroxyclerod-3,13-dien-15,16-olide (CD, Figure 3), a clerodane diterpene isolated from *Polyatlthia longifoliav* var. *pendula* (Annonaceae), has been reported to have various biological activities, including anti-inflammatory and antitumor properties [213,214]. Liu et al. [100] studied the effects of CD on human clear RCC lines 786-O and A-498 and its cytotoxic and apoptotic mechanisms. The results of this study found that CD caused cell death through increased ROS production and increased mitochondria-dependent apoptosis in clear cell RCC cells. It was also found that CD blocked Akt/mTOR and MEK/ERK1/2 pathways and their downstream molecules, such as HIF-2α, c-Myc, and FOXO3a. All of the results suggest that CD has potential for use as a chemotherapeutic agent for the treatment of RCC.

#### 3.2.17. Kahweol

Kahweol (Figure 3), a diterpene molecule derived from coffee beans, has shown anti-inflammatory, anti-osteoclastogenesis, and anticancer effects [198,215]. Um et al. [101] studied the antitumor effects of kahweol in combination with TRAIL using renal carcinoma Caki cells. Kahweol in conjunction with TRAIL induced PARP cleavage, caspase-8 activation and the downregulation of NF-κB p65 subunit, c-Flip, and Mcl-1 proteins, suggesting that kahweol potentiated TRAIL-induced apoptosis in Caki cells. Min et al. [102] explored kahweol’s ability to enhance sorafenib-mediated apoptosis in Caki cells. Kahweol combined with sorafenib induced accumulation of a sub-G1 population and PARP cleavage. Further treatment with this combination increased caspase-3 activation and downregulated c-Flip and Mcl-1 expression. Additionally, A498 and ACHN cells treated with this combination showed increased levels of PARP cleavage and downregulated c-Flip and Mcl-1 expression.

#### 3.2.18. Lycopene

Lycopene (Figure 3), a carotenoid found in various fruits and vegetables, including most notably the tomato, has been shown to have anticancer and antioxidant activity [216,217]. Sahin et al. [135] evaluated the effects of lycopene on the growth of renal cancer in female Ekar rats with TSC2 mutation, a gene that has been evidenced to cause renal carcinogenesis in this model. The product of TSC2 gene, tuberin, normally operates to regulate cell reproduction and growth through negative regulation of the mTOR pathway. Lycopene was supplemented at a low (100 mg/kg of diet) or high (200 mg/kg of diet) dosage over a period of 18 months. Lycopene administration significantly decreased the development and growth of renal tumors. Immunohistochemical analysis of tumor samples revealed no difference in the expression of mTOR, phospho-s6, and EGFR compared to the control group.

#### 3.2.19. Nimbolide

Nimbolide (Figure 3), a major limonoid constituent present in the leaves and flowers of neem (*Azadirachta indica*), has been shown to exhibit potent antiproliferative effects against various cancer cell lines and chemotherapeutic efficacy in animal tumor models [218,219]. In a study performed on 786-O and A-498 RCC cells, nimbolide treatment demonstrated increased cytotoxicity with increased concentrations of up to 6 μM compared to human normal renal tubular cells (HK2 cells), which revealed an absence of increased cytotoxicity for concentrations below 6 μM. Furthermore, nimbolide induced cell cycle arrest at the G2/M phase through increased levels of p-p53, p-cdc2, and p-cdc25c, DNA damage, and decreased levels of cyclin A, cyclin B, cdc2, and cdc25c. Nimbolide treatment also induced apoptosis through increased levels of γ-H2AX, cleaved-caspase-3, cleaved-PARP, Bax, DR5, and CHOP and decreased levels of pro-caspase-8, MCL-1, and Bcl-2 [103].

#### 3.2.20. Oridonin

Oridonin (7,20-epoxy-*ent*-kaurane, Figure 3), a diterpenoid extracted from the traditional Chinese herb *Rabdosia rubescens*, has been shown to have antifibrotic, anti-inflammatory, antibacterial, and anticancer effects [220,221]. Zheng et al. [104] studied the antiproliferative effects of oridonin on 786-O cancer cells. Exposure to oridonin induced a stronger cytotoxic response in the 786-O cells compared to normal renal podocyte cells. Oridonin induced activation of apoptotic signaling pathways, as shown by increased protein levels of cleaved caspase-3, Bax, receptor interacting protein-3 (RIP-3), phosphorylated extracellular signal-regulated kinase (p-ERK), and p-p38, elevated release of LDH and high mobility group box 1 protein (HGMB1), increased activity of PARP-1, and decreased levels of Bcl-2. Oridonin induced necroptosis by increasing levels of RIP-1 and RIP-3. Oridonin also enhanced 5-fluorouracil (5-FU) cytotoxicity of 786-O cells through depletion of glutathione (GSH) and increased p-JNK, p-ERK, and p38 levels.

Zheng et al. [104] extended their in vitro study to determine the antitumor efficacy of oridonin in nude mice xenografted with 786-O cells. The experimental animals were treated with oridonin (20 mg/kg, every three days) alone, 5-FU (25 mg/kg, every three days) alone, or a combination of both agents (20 mg/kg of oridonin plus 12.5 mg/kg of 5-FU). The dual therapy group was the most effective in lowering the tumor weights compared to the other groups, indicating that oridonin enhanced the antitumorigenic effect of 5-FU. Additionally, it was found that oridonin increased levels of HMG1, Bax, cleaved caspase-3, RIP-1, RIP-3, activity of Parp-1, and decreased levels of Bcl-2, suggesting it was able to induce apoptosis and necrosis in this mouse xenograft model.

#### 3.2.21. Parthenolide

Parthenolide (Figure 3), a sesquiterpene lactone present in the traditional medical plant feverfew (*Tanacetum parthenium*), has exhibited various biological and pharmacological properties, such as anti-inflammatory, redox-modulating, epigenetic and anticancer effects [222,223]. Oka et al. [105] investigated parthenolide’s cytotoxic effect on renal cancer by using the OUR-10 and ACHN cell lines. Parthenolide was shown to inhibit cell proliferation and induce apoptosis compared to the control cells. Parthenolide-treated cells exhibited decreased expression of NF-κB, p-NF-κB, and p65, reduced nuclear staining of NF-κB p65, and increased IκBa levels. These results suggest that parthenolide inhibits proliferation and induces apoptosis through inhibition of the NF-κB pathway.

Oka et al. [105] also evaluated the antitumor effectiveness of parthenolide in OUR-10 tumor xenografts in mice. Oral administration of 10 mg/kg daily of parthenolide was tested, and it was found that in parthenolide-treated mice the mean tumor volume was significantly less than that in the control mice. The animals injected with parthenolide (3 μg/mouse, 3 times per week) were found to have significantly decreased tumor volumes compared to the controls. In addition, NF-κB, Bcl-xL, COX-2, IκBα, and MMP-9 levels in the tumor tissue of parthenolide-injected animals were decreased compared to their levels in the untreated control.

#### 3.2.22. Silibinin

Silibinin (Figure 3), a flavonoid derived from milk thistle (*Silybum marianum*), has been found to exhibit encouraging antioxidant and anti-inflammatory properties [224,225]. Chang et al. [106] found that silibinin did not induce a significant decrease in cell viability in 786-O cells. However, it was observed that silibinin induced the inhibition of cell migration and invasion. Exposure to silibinin led to suppressed phosphorylation of ERK1/2 and p38 and decreased levels of NF-κB, c-Jun, and c-Fos. Silibinin-treated cells also showed reduced mRNA levels of MMP-2, MMP-9, and urokinase plasminogen activator (u-PA).

Chang et al. [106] extended their in vitro study as above to investigate silibinin’s effect on nude mice inoculated with 786-O cells. Silibinin treatment of either 100 or 200 mg/kg was given orally daily for 44 days. Morphometric analysis revealed that silibinin decreased average tumor volume by 70.1% and decreased tumor weight by 69.7% in the group that received a maximum dose of silibinin. It was also found that the cytotoxicity of either paclitaxel or 5-FU was enhanced in the presence of silibinin. However, silibinin did not enhance the cytotoxicity of everolimus or of the combination of everolimus and 5-FU.

#### 3.2.23. Sorghumol

Sorghumol (Figure 3), a triterpene derived from the roots of rasana (*Pluchea lanceolata*), was first isolated by Wang et al. [226]. There are very limited studies on the anticancer potential of this compound. Li et al. [107] explored the anticancer activity of sorghumal in renal cancer. The study revealed that treatment of A498 renal cells with sorghumol decreased cell growth and proliferation through increased apoptosis, G2/M cell cycle arrest, and downregulation of p-mTOR, p-PI3K, and p-Akt expression.

#### 3.2.24. Thymoquinone

Thymoquinone (Figure 3), a monoterpene isolated from black seed (*Nigella sativa*), was found to have anti-inflammatory, antioxidant, and anticancer activity [227,228]. Park et al. [108] set out to explore thymoquinone’s anticancer effect on renal cancer using Caki cells. Thymoquinone induced apoptosis through induction sub-G1 population accumulation and downregulation of c-FLIP and Bcl-2 expression. Thymoquinone also induced ROS generation, which led to reduced levels of mitochondrial membrane potential and cyt. c release. It was also found that thymoquinone was effective in inducing apoptosis in other renal cancer cells, such as ACHN and A498 cells. In another study performed by Liou et al. [109], thymoquinone did not exhibit significant cytotoxic activity against 786-O-SI3 RCC cells or non-malignant HK-2 renal cells. Thymoquinone, however, showed antimetastatic activity against 786-O-SI3 cells. Exposure to thymoquinone depressed cell transmigration and invasion through inhibition of translation, transcription, and proteolytic activity of MMP-2 and u-PA, cell adhesion to type I and type IV collagen, and TGF-β1 promoted cell motility. Additionally, thymoquinone suppressed the PI3K/Akt and Src/paxillin signaling cascade through downregulation of p-phosphatidylinositol 3-kinase, p-Akt, p-Src, p-paxillin, fibronectin, N-cadherin, and Rho A expression.

Liou et al. [109] also studied the efficacy of thymoquinone against the 786-O-SI3 xenograft tumor model in male C57BL/6 mice. The 786-O-SI3 cells were transfected with a luciferase expressing vector, allowing for measurement of metastasis based on quantifying luciferase activity with a sensitive light imaging system. It was found that thymoquinone, when administered at 10 or 20 mg/kg orally per day for 42 days, was effective in reducing lung metastasis of 786-O-SI3 cells. Histological examination of the lung samples demonstrated an attenuated number of 786-O-SI3 cells in lung tissues, further demonstrating thymoquinone’s its ability to inhibit metastasis.

#### 3.2.25. Tonantzitlolone

Tonantzitlolone (Figure 3) is a diterpene derived from the native Mexican plant *Stillingia sanguinolenta* Mull. Arg. [229,230]. Tonantzitlolone has been known to have antiproliferative properties against certain cancer cell lines and possess antitumor activity [229,230]. Sourbier et al. [110] investigated cytotoxicity targets and mechanisms of action of tonantzitlolone in clear cell RCC. It was found that tonantzitlolone exhibited anticancer activity by altering metabolic pathways in cancer cells. The results of this study showed tonantzitlolone activated protein kinase C-α (PKCα) and protein kinase C-θ (PKCθ), which induced an insulin-resistant phenotype by inhibiting IRS1 and the PI3k/Akt pathway. Additionally, tonantzitlolone activated heat shock factor 1 (HSF1), which amplifies the glucose dependency of cancer cells. Rubaiy et al. [111] investigated if tonantzitlolone would activate TRPC1/4/5 channels. From this study, it was found that tonantzitlolone elevated intracellular calcium in A498 cells, exhibiting its effects as an agonist for TRPC1/4/5 channels, achieving selective cytotoxicity against human renal carcinoma cells.

#### 3.2.26. Triptolide

Triptolide (Figure 3), a diterpenoid triepoxide derived from the Chinese medicinal herb thunder god vine (*Tripterygium wilfordii* Hook F), has been used for its anti-inflammatory, immunosuppressive, and antitumor activities [231,232,233]. Triptolide was investigated for its potential anticancer effect on both 786-O and OS-RC-2 cells by Li et al. [68]. Triptolide induced apoptosis accompanied by increased levels of Bax and decreased levels of Bcl-2 and Bcl-xL. Triptolide was also found to induce cell cycle arrest at S phase through reduced levels of Rb, A/CDK1, CDK2, and B/CDK. In another study, the antitumor effects of triptolide in combination with TRAIL were studied in A498, Caki-1, ACHN, 786-O, and 769-P cells. Each cell line demonstrated a concentration-dependent increase in TRAIL-induced apoptosis when treated with increasing concentrations of triptolide. In ACHN cells, triptolide increased susceptibility to TRAIL-induced apoptosis by induction of TRAIL-R2 (DR5) expression and downregulation of heat shock protein 70 (HSP70) expression [112].

#### 3.2.27. Triptolidenol

Triptolidenol (Figure 3), an epoxy diterpene lactone isolated from thunder god vine (*Tripterygium wilfordii*), has been shown to possess anticancer and anti-inflammatory properties; however, the detailed mechanisms have not been previously studied [113]. Jin et al. [234] investigated anticancer effects and molecular mechanisms of action of triptolidenol against a panel of renal cancer cells. Triptolidenol was found to reduce cell proliferation in 786-O and ACHN cells by inducing apoptosis through increased levels of cytochrome c, cleaved caspase-3, cleaved caspase-9, cleaved PARP, and Bax and decreased levels of Bcl-2 and mitochondrial membrane potential. Triptolidenol also induced S phase cell cycle arrest through decreased levels of cyclin A2, cyclin D1, cyclin E1, CDK2, and CDK4. Furthermore, it suppressed COX-2/NF-κB signaling through attenuation of NF-κB kinase and inhibitor of nuclear factor -κB kinase subunit β (IKKβ) activity.

#### 3.2.28. Zerumbone

Zerumbone (Figure 3), a cyclic 11-membered crystalline sesquiterpene compound isolated from the rhizomes of ginger (*Zingiber zerumbet*), has been shown to exert diverse biological and pharmacological effects, such as antioxidant, anti-allergic, immunomodulatory, and anticancer activities [234,235]. To determine its effects on renal cancer, 786-O, 769-P, and Caki-1 cell lines were used. Zerumbone exhibited inhibition of cell viability and induced apoptosis in 786-O and 769-P cells with an increase in protease activity of caspase-3 and caspase-9, an increase in cleavage of PARP, and suppression of Gli-1 and Bcl-2 [114]. In a similar study, it was found that zerumbone exerted its anticancer effects in 786-O and Caki-1 cells through suppression of colony formation. Zerumbone also inhibited the STAT3 signaling pathway by decreased levels of c-Src and Janus kinase 1 (JAK1), activation of JAK2, phosphorylation of STAT3, and reduced DNA binding activity of STAT-3. Additionally, zerumbone induced expression of Src homology 2 domain-containing protein tyrosine phosphatase-1 (SHP-1), caspase-3, and PARP, and suppressed expression of cyclin D1, Bcl-2, Bcl-xL, Mcl-l, survivin, MMP-9, VEGF, Ki-67 and CD31 [115].

Zerumbone was studied by Shanmugam et al. [115] for its antitumor potential in athymic mice bearing 786-O tumor xenografts. Intraperitoneal administration of zerumbone treatment at 50 mg/kg 5 times a week for 6 weeks was effective in causing significant suppression of tumor growth compared to the controls. In further evaluation, it was found that zerumbone downregulated expression of Ki-67, CD-31, Bcl-2, and p-STAT3, and increased expression of SHP-1 mRNA and SHP-1 in tumor tissues collected from the experimental animals.

### 3.3. Alkaloids

#### 3.3.1. (-)-Antofine

(-)-Antofine (Figure 4), a phenanthroindolizidine alkaloid, is extracted from the plants of the Asclepiadaceae family, such as *Tylophora*, *Cynanchum*, and *Pergularia*. (-)-Antofine has previously exhibited antiviral, antimicrobial, and anticancer activities [236,237]. In a study performed by Song et al. [116], (-)-antofine was found to decrease cell growth in Caki-1 (clear cell RCC) cell lines. (-)-Antofine increased MET degradation and prevented MET endocytosis. Additionally, (-)-antofine decreased cell growth via reduced expression of MET, STAT3, growth-factor receptor bound protein-2 (Grb2), Ras-related C3 botulinum toxin substrate 1 (RAC1), non-receptor tyrosine kinase (Src), and ERK1.

Song et al. [116] extended their in vitro study to investigate the effects of (-)-antofine against a Caki-1 tumor xenograft model in vivo. (-)-Antofine decreased STAT3 activation and reduced the translocation of STAT3, leading to the inhibition of cancer cell migration. (-)-Antofine at a dose of 5 mg/kg reduced tumor growth and decreased Met-mediated STAT3.

#### 3.3.2. Berberine

Berberine (Figure 4), an isoquinoline alkaloid extracted from various medicinal plants, including barberry (*Berberis vulgaris*), Oregon grape (*Berberis aquifolium*), and Indian barberry (*Berberis aristata*), has various pharmacological effects, such as anti-inflammatory, antimicrobial, and antitumor activities [238,239,240]. According to a study conducted by Lee et al. [117], berberine was found to induce apoptotic cell death in Caki (RCC) cells. It exhibited downregulation of two proteins, c-FLIP and Mcl-1. Furthermore, berberine caused an increase in TRAIL-induced apoptosis via ROS mediation. Lopes et al. [118] found that berberine decreased cell viability and increased cell death in ACHN and 786-O RCC cells. Before berberine was used with photodynamic therapy (PDT), berberine-treated cells showed good viability, demonstrating that berberine has relatively low cytotoxicity. The PDT was used as a noninvasive treatment method using a light source that could be visualized, photosensitizing agent, and oxygen. Altogether, the environment creates a photochemical reaction, producing ROS and leading to damage of the cancer cells [241,242]. When berberine and PDT were used together, there was a significant reduction in cell viability. Treated ACHN and 786-O RCC cells showed an increase in ROS, autophagy resulting in cell death, and caspase-3 leading to cell death. c-FOS-induced growth factor (fIGF) and telomerase reverse transcriptase (TERT) decreased in expression and polo like kinase 3 (PLK3) increased when treated with berberine and PDT. Cells that were treated with berberine and PDT also showed significant increases or decreases in their metabolites. Lysine significantly increased, while lactate and formate both significantly decreased [118].

#### 3.3.3. Cepharanthine

Cepharanthine (Figure 4), a biscoclaurine alkaloid from the plant *Stephania cepharantha* Hayata, has exhibited biological immunoregulatory, anti-inflammatory, antioxidant, antiparasitic, and anticancer properties [243,244,245]. In a study performed by Shahriyar et al. [119], cepharanthine was found to decrease cell viability. Cepharanthine increased UPS53 and DR5 and specifically decreased survivin in A498 cells (RCC). Cepharanthine and TRAIL combined showed increased sub-G1 population and PARP-1 cleavage. Furthermore, treatment induced caspase-3 activation, resulting in decreased cell viability through TRAIL-induced apoptosis in Caki-1 and ACHN cell lines. Decreased cFLIP and STAM binding protein like 1 (STAMBPL1) were observed after treatment as well.

#### 3.3.4. Chelerythrine

Chelerythrine (Figure 4), a natural benzo[c]phenathridine alkaloid, can be extracted from various plant species, such as greater celandine (*Chelidonium majus*), the five-seeded plume-poppy (*Macleaya cordata*), and bloodroot (*Sanguinaria canadensis*). Chelerythrine has been shown to inhibit proliferation and promote apoptosis in various cancer cell lines [246,247,248]. In a study exploring its effect on RCC, chelerythrine was found to reduce cell viability. The proposed mechanisms were through the regulation of Bcl-2 family protein expression and activation of the mitochondrial pathway. Chelerythrine induced cell apoptosis at the G2/M phase and activated the ER stress pathway through ROS. Additionally, the study demonstrated that chelerythrine could kill human renal cancer cells, but spare healthy cells [120].

#### 3.3.5. Dauricine

Dauricine (Figure 4), a bisbenzylisoquinoline alkaloid isolated from the rhizome of moonseed (*Menispermum dauricum DC*), has displayed antiarrhythmic, anti-inflammatory, and anticancer properties [249,250,251]. Dauricine was found to inhibit cell growth and cell viability in 786-O, A498, ACHN, and Caki-1 cells. Mechanistically, dauricine caused a decrease in cyclin D1, CDK4, CDK2, pro-caspase-9, Bcl-2, and MCL-1, and increase in p21 and Bax levels. This resulted in cell cycle arrest at the G0/G1 phase, induction of apoptosis through the intrinsic pathway, and inhibition of the PI3K/Akt signaling pathway [121].

#### 3.3.6. Neferine

Neferine (Figure 4), a bisbenzylisoquinoline alkaloid extracted from the seed embryo of the lotus plant (*Nelumbo nucifera*), is reported to possess antioxidant, anti-inflammatory and anticancer activities [252,253,254,255]. The anti-renal cancer potential of neferine has been studied using Caki-1, ACHN, and A498 cells. NF-κB is responsible in the cell for regulating Bcl-2 gene transcription. The inhibition of the pathway resulted in apoptosis in RCC cells. Neferine caused inhibition of cell proliferation and induced apoptosis through decreased cell viability by lowering levels of Bcl-2, which promoted apoptosis of cancer cells. Specifically, it was shown that the NF-κB pathway was inhibited and there was an increase in p65, which is a subunit of the NF-κB pathway in regulation of pro- or anti-apoptotic signals [122].

#### 3.3.7. Neopapillarine

Neopapillarine (Figure 4) is a counarino-alkaloid from the root extract of *Neocryptodiscus papillaris*. It has many active biological properties, such as antioxidant, anti-inflammatory, antiviral, antibacterial, antifungal and anticancer activities [256,257,258]. Tosun et al. [123] found that neopapillarine decreased cell growth in A498 and UO31 RCC cell lines, although mechanisms were not specified.

#### 3.3.8. Oxymatrine

Oxymatrine (Figure 4), a quinolizidine alkaloid extracted from the shrubby sophora (*Sophaora flavescens*), has various pharmacological properties, such as anticancer, antifibrotic, antiviral, antiallergic, anti-inflammatory, antiarrhythmic, and cardiovascular protective effects [259,260,261,262]. Oxymatrine was tested using A498 and SW839 cell lines and demonstrated decreased cell viability and proliferation and increased cell death, as well as sensitivity to other chemotherapeutic drugs such as taxol. Both cells exposed to oxymatrine showed significant decreases in Ki-67 expression with cells in the G1 phase. It has was also observed that oxymatrine arrested cells in the G1 phase along with downregulation of cyclin D1. Other important regulators in the cell cycle with altered levels after treatment with oxymatrine included reduced levels of CDK6 and increased levels of p27. The levels of cleaved caspase-3 and cleaved PARP were increased in the cells as well. Oxymatrine significantly decreased MMP-9, MMP-2, and vimentin and increased the E-cadherin level, demonstrating its ability to stop migration and invasion of the cancer cells. Furthermore, there was a decrease in β-catenin translocation. As for combined treatment, taxol and oxymatrine aided in taxol-modulated inhibition of cell viability. Furthermore, taxol and oxymatrine increased taxol-induced apoptosis, supporting the pairing of a chemotherapeutic drug with a phytochemical. An increase in E-cadherin and glycogen synthase kinase 3β (GSK-3β) was reported [124].

Oxymatrine was additionally tested in vivo using BALB/c nude mice xenografted with A498 or SW839 cells. Treatment with oxymatrine (50 mg/kg) via intraperitoneal injection for 28 days suppressed tumor cell growth by increasing apoptosis and decreasing β-catenin [124].

#### 3.3.9. Piperlongumine

Piperlongumine (Figure 4), an alkaloid found in long pepper (*Piper longum*), has been implicated in suppressing the growth of numerous cancer cells by pleiotropically modulating different oncogenic signaling pathways [263,264,265]. Makhov et al. [125] studied the effects of piperlongumine on the 786-O RCC cell line, where it was shown to decrease cell viability and proliferation. Treatment with piperlongumine caused a significant reduction in Akt/mTOR signaling. The levels of Akt downstream effectors, TSC2 and GSK-3β, were significantly decreased in the 786-O cell line. Inhibition of Akt via piperlongumine administration resulted in a notable decrease in mTORC1 complex activity, represented by decreased phosphorylated levels of mTORC1 effectors. Furthermore, there was an elevated ROS production and induction of autophagy (based on LC3-II protein accumulation) in cells treated with piperlongumine in a concentration-dependent manner. Furthermore, the study found that piperlongumine reduced levels of Unc-51 like autophagy activating kinase (ULK1, a mammalian autophagy-initiating kinase) and induced cell death in the 786-O cell line. In another study by the same group, the anticancer effect of piperlongumine was tested in RCC lines 786-O and PNX0010. It reduced cell viability and inhibited cell growth. Piperlongumine inhibited c-Met, also known as tyrosine protein kinase Met or hepatocyte growth factor receptor, as well as the expression and downstream signaling of ERK/MAPK, STAT3, NF-κB and Akt/mTOR via the ROS-dependent mechanisms [126].

Golovine et al. [126] further investigated piperlongumine’s effects in vivo using xenografted PNX0010 cells in C.B17/lcr-scid 6-week-old male mice. The mice were given piperlongumine, piperlongumine-dimer, or piperlongumine-fluorophenyl at 20 mg/kg intraperitoneally 3 times/week for 18 days. Overall, piperlongumine decreased c-Met, resulting in decreased tumor growth and decreased cell viability. Piperlongumine-dimer and piperlongumine-fluorophenyl proved to have better results and even enhanced the results already listed. More importantly, both piperlongumine-dimer and piperlongumine-fluorophenyl effects were established at a lower dose than that of piperlongumine alone.

#### 3.3.10. Tetrandrine

Tetrandrine (Figure 4), a bisbenzylisoquinoline alkaloid derived from the roots of a medicinal plant, *Stephania tetrandra*, has exhibited promising antineoplastic activities in tumor models [266,267,268]. Tetrandrine demonstrated inhibition of growth and angiogenesis, promotion of apoptosis, and cell cycle arrest in 786-O and 769-P cell lines. These results were mediated through a decrease in levels of p-Akt, p-PI3K, p-PDK1, NF-κB, and MMP-9 in the Akt/NF-κB/MMP-9 signaling pathway [127].

### 3.4. Sulfur-Containing Compounds

#### 3.4.1. Allicin

Allicin (Figure 5), a major component of garlic (*Allium sativum* L.), is one of the earliest plants used medicinally due to its therapeutic properties [269,270]. It is also commonly found in shallots, leaks, onions, and chives [271]. Allicin has previously been shown to have antimicrobial, antiarthritic, and antitumor properties [128,269,270,271]. Song et al. [128] found that allicin decreased cell viability, cell differentiation, and cell chemotactic ability in RCC-9863 cell lines. Additionally, allicin downregulated Bcl-2, increased Bax, and decreased VEGF via inhibition of HIF-1α.

#### 3.4.2. Sulforaphane

Sulforaphane (Figure 5) is a natural compound present in cruciferous vegetables, including cabbage and broccoli [272]. It has demonstrated antiproliferative activity against cancer cell lines and inhibited tumor growth in animal models [273,274]. Everolimus is a chemotherapeutic drug, and during treatment, patients could develop resistance against this medication. Juengel et al. [129] used sulforaphane to investigate its effect against everolimus-resistant (Caki-1^res^, KTCTL-26^res^, and A498^res^) and everolimus-sensitive (Caki-1^par^, KTCTL-26^par^, and A498^par^) kidney cancer cells. Sulforaphane reduced the growth of both Caki-1^res^ and Caki-1^par^ cell lines; however, the effect was more pronounced in Caki-1^par^ cell line. Furthermore, this finding was correlated to an increase in G2/M- and S-phase cells. Sulforaphane led to a reduction in integrins α5, α6, β1, and β4, with a more drastic effect in Caki-1^par^ cells compared to Caki-1^res^cells. In both Caki-1^par^ and Caki-1^res^ cells, Cdk1, p-Cdk2, Cdk2, cyclin A, cyclin B, and p27 were downregulated after treatment with sulforaphane, while p19 was upregulated. In Caki-1^res^ cells, p-Akt and p-Rictor were downregulated. Furthermore, in Caki-1^res^ cells, adhesion and chemotaxis were diminished by treatment with sulforaphane, while adhesion was enhanced in Caki-1^par^ cells. Based on these results, sulforaphane may have potential for treating RCC patients with everolimus resistance. In an additional study, Juengel et al. [130] used RCC cell lines, namely Caki-1, KTCTL-26, and A498, to investigate sulforaphane’s ability to prevent resistance in everolimus-treated cells. When sulforaphane was used in combination with everolimus, there was a profound decrease in cell growth. Sulforaphane was found to lead to a G2/M phase arrest with short-term treatment (24 h) and a G0/G1 phase arrest with long-term treatment (8 wk). Short-term treatment with sulforaphane alone upregulated cdk1, cdk2, cyclin A, and cyclin B. Long-term treatment with sulforaphane downregulated cdk1 and cdk2. Combination treatment reduced cyclin A and cyclin B both in the short-term and long-term. Sulforaphane was also found to downregulate p-Akt and p-Raptor levels to a significant extent in everolimus-sensitive cells with both durations of treatment. Interestingly, sulforaphane (8 wk) used alone blocked RCC cell line growth more efficiently than combination treatment. Based on the results, it seems likely that sulforaphane may not be able to completely prevent resistance to everolimus treatment, but it may delay the development of resistance.

## 4. Phytocompounds in Renal Cancer Research: Clinical Studies

There are a multitude of epidemiologic studies that suggest the consumption of vegetables and fruits can improve overall health and decrease the risk of a variety of cancers. Many observational studies with retrospective and prospective designs have been conducted to determine the association between diet and the risk of developing RCC. The impact of individual phytochemicals on RCC in randomized controlled trials is limited to a few studies. In the following section, we present studies incorporating dietary agents and specific phytochemicals in renal cancer.

Yuan et al. [275] conducted a case–control study involving 1276 patients with confirmed RCC and determined if there was an association between cruciferous vegetable consumption and risk of RCC development. The highest quintile for cruciferous vegetable consumption versus the lowest quintile showed a significant inverse association with the risk of RCC (OR = 0.53, *p* < 0.001). The investigators identified an inverse association between consumption of carotenoids and risk of RCC for the highest quintile versus the lowest quintile for α-carotene (OR = 0.61), β-carotene (OR = 0.69), lutein (OR = 0.70), and β-cryptoxanthin (OR = 0.73). Furthermore, the inverse association between cruciferous vegetable consumption and risk of RCC was stronger than that of any of the nutrient–RCC associations; therefore, other potentially unknown compounds present in cruciferous vegetables may have a protective effect against RCC (Table 3). Lee et al. [276] performed a prospective cohort study involving 88,759 women in the Nurses’ Health Study and 47,828 men in the Health Professionals Follow-up Study with 248 developing RCC after 4 years. Dietary intake of fruits and vegetables was associated with a decreased risk of RCC in men (RR = 0.45, CI = 0.25–0.81, of ≥6 servings of fruit and vegetable intake/day versus <3 servings/day), but not in women. For the highest quintile of intake versus the lowest, there was a significant inverse association between consumption of total carotenoids (RR = 0.83, CI = 0.71), vitamin A (RR = 0.47, CI = 0.26–0.84) and vitamin C (RR = 0.51, CI = 0.30–0.88) and risk of RCC in men. Lee et al. [277] corroborated the results regarding the inverse association of fruit and vegetable consumption in an extension of their previous study (RR = 0.68, CI = 0.54 for ≥600 g/d compared to <200 g/d) and risk of RCC. In this pooled analysis of 13 prospective studies of 774,952 participants, dietary intake of α-carotene (RR = 0.82, CI = 0.69–0.98) showed a significant inverse association with the risk of RCC. However, no significant association was identified with the consumption of β-carotene, β-cryptoxanthin, or lutein/zeaxanthin. Additionally, this study did not verify a difference in risk between males and females.

The risk of RCC associated with overall dietary habits has been analyzed in numerous studies. Bosetti et al. [278] conducted a case–control study in Italy on 767 hospitalized patients with histologically confirmed RCC. In this study, these investigators identified a significant inverse association between dietary vitamin E intake and a borderline significant inverse association of vitamin C intake and risk of RCC. Among these same participants, the investigators conducted a separate study as an extension of the previous study to investigate the association of RCC development and total flavonoid consumption [279]. This group was the first to identify an inverse association of the intake of flavones (OR = 0.68, CI = 0.50–0.93 with the odds ratios for subjects in the highest versus the lowest quintile of intake)) and flavonols (OR = 0.69, CI = 0.50–0.95) with RCC. In contrast to Bosetti et al. [278], a Canadian case–control study of 1138 patients with confirmed RCC found no clear association between dietary intake of vitamin C and E, β-cryptoxanthin, and lycopene and risk of RCC [280]. Nevertheless, this study did confirm a significant inverse association for the dietary intake of β-carotene (OR = 0.74 CI = 0.59–0.92 for the highest vs. lowest quartile) and lutein/zeaxanthin (OR = 0.77, CI = 0.62–0.95), as previously reported by Yuan et al. [275]. Furthermore, this association was found to be more pronounced in women and obese individuals [280].

Brock et al. [281] conducted a case–control study among 323 Iowa residents between the ages of 40–85 with confirmed RCC. This group was the first to identify a significant inverse association between vegetable fiber (OR = 0.4, CI = 0.2–0.6) and risk of RCC. In contrast, this association was not observed for consumption of fruit fiber or grain fiber. Additionally, β-cryptoxanthin was the only micronutrient with a significant inverse association (OR = 0.6, CI = 0.3–0.9).

Bock et al. [282] conducted a case–control study involving 1142 confirmed cases of RCC in Iowa among European American and African American residents. These investigators validated the results of some of the previous studies mentioned. A significant inverse association between the consumption of α-carotene, β-carotene, lutein zeaxanthin, vitamin A, vitamin C, vitamin E, and β-cryptoxanthin and risk of RCC was observed. Additionally, this inverse association was identified for the consumption of lycopene, selenium, folate, and thiamine. However, these findings did not differ among race, gender, age, or smoking status. Ho et al. [283] conducted a prospective cohort study among 96,196 postmenopausal women aged 50–79. This study supported the inverse association between lycopene intake (HR = 0.61, CI = 0.39–0.97 with the highest quartile versus the lowest quartile) and risk of developing RCC. However, these investigators did not find an association with intake of α-carotene, β-carotene, β-cryptoxanthin, lutein with zeaxanthin, vitamin C, or vitamin E.

Berotoia et al. [284] conducted a prospective cohort study in Finland consisting of 27,062 male smokers aged 50–69. In addition to tracking their daily dietary habits, each participant was treated with daily supplementation of β-carotene (20 mg p.o.) and/or vitamin E (50 mg) and followed up for 19 years. These investigators did not identify any significant associations between the intake of fruit, vegetables, or antioxidants and the risk of RCC.

Unfortunately, there are limited studies examining the effects of individual phytochemical consumption on the risk of RCC. Marshall et al. [285] conducted a pilot study on 45 patients with metastatic RCC. These patients ingested coumarin (100 mg daily) and cimetidine (300 mg, 4 times/day on day 15). Coumarin has previously shown antitumor properties, and cimetidine is incorporated as it has demonstrated immunomodulatory properties. Three patients showed complete remission, 11 patients had partial remission, and 12 patients experienced stabilization of disease. Herrmann et al. [286] conducted a similar phase II trial in Berlin, Germany, involving 31 patients with confirmed RCC. These patients ingested coumarin (100 mg daily) and cimetidine (400 mg, 4 times/day on day 15). The exceptional results obtained by Marshall et al. [285] were not confirmed in this study. Two patients achieved partial responses, and this response was observed through up to 73 weeks of follow-up. One other patient had a mixed response with >50% reduction in metastasis in his right lung with the concurrent development of new lesions in his left lung. Five patients developed stable disease for 26+, 28, 30, 31, and 45+ weeks [286]. Dexeus et al. [287] conducted a study utilizing the same treatment methods and achieved response rates similar to those achieved by Herrmann et al. [286]. Three patients (6%) achieved a partial response, one of which was after a dose escalation. The dose escalation was 100 mg coumarin 4 times/day and was implemented when the disease progressed. Kokron et al. [288] conducted a pilot study in Austria among 38 patients with metastatic RCC and one patient with a second primary RCC. Patients ingested cimetidine (400 mg daily) followed by coumarin (100 mg daily) after 1 week. Overall, two patients achieved a complete response (30 and 50+ months), and 3 patients achieved a partial response (8, 13, and 14 months) with a total response rate of 12.8%. Sagaster et al. [289] conducted a randomized phase III trial in Austria involving 148 patients with metastatic RCC. The patients were randomized to receive either interferon-α (IFN-α) (5 million units five times/week, s.c.) with coumarin (100 mg/d, p.o.) and cimetidine (3 × 400 mg/d, p.o.), or IFN-α monotherapy (5 million units five times/week, s.c). No difference in response rates (RR = 17.1% for IFN-α + coumarin + cimetidine, RR = 20.8% for IFN-α) or overall survival time was observed between the groups.

Although there are limited studies investigating the effects of individual phytochemical consumption on the risk of RCC, the research that has been conducted strongly suggests that dietary fruit and vegetable consumption reduces the risk of developing RCC. This reduced risk may be correlated with the quantity of known phytochemicals or may be due to the actions of unidentified compounds. Further studies are necessary to identify these other constituents and determine their benefits and mechanisms of action. Furthermore, many of these listed studies had contradictory results; therefore, additional studies are necessary to adequately quantify the effects of these individual phytochemicals.

## 5. Conclusions and Future Perspectives

Kidney cancer remains one of the deadliest cancers, especially when metastases are developed. Conventional chemotherapeutic drug therapy has shown low success against RCC. There is an apparent need for novel therapeutics for the prevention and treatment of RCC. Previous reviews on RCC and phytochemical therapy involved fewer phytochemicals, did not include mechanisms of action, or did not include all types of studies, such as in vitro, in vivo, and clinical trials. This review encompassed a complete review of phytochemicals with all study types for potential use in the prevention and treatment of RCC. In this review, we evaluated the therapeutic potential of phytochemicals in RCC with an understanding of mechanisms of action based on available literature.

Phytochemicals have therapeutic potential against renal cancer as evident from a meticulous examination of both preclinical (in vitro and in vivo) and clinical studies. A total of 114 individual publications were included in our systematic review (Figure 1). All of these studies utilized four main groups of phytochemicals, namely phenolics, terpenoids, alkaloids, and sulfur-containing compounds. Phenolics comprised 43% of the in vitro studies and 54% of the in vivo studies. Terpenoids made up 41% of the in vitro studies and 36% of the in vivo studies. Alkaloids accounted for 13% of the in vitro studies and 11% of the in vivo studies. Sulfur-containing compounds accounted for only 3% of the in vitro studies with no reports on in vivo studies (Figure 1).

The literature evaluated in this article displayed several cellular and molecular mechanisms of action affecting various signaling pathways implicated in RCC. Various cancer hallmarks and events as well as associated signaling pathways and targets affected included oxidative stress (ROS, GSH, SOD, and GST), inflammation (IL-1β, IL-6, IL-12, and TNF-α), proliferation (c-Myc, β-catenin, and mTOR), cell cycle arrest (cyclin A, cyclin D1, cdk4, p21, and p53), cell death (Bax, Bak, and caspase-3), angiogenesis (VEGF and HIF-1α), invasion and migration (MMP-2, E-cadherin, Snail1, and ZEB2), and metastasis (CXCL4, Twist, ZEB1, and Vimentin) (Figure 6). In clinical studies, dietary intake of natural compounds, such as carotenoids, flavonoids, and vitamins, was associated with a decreased risk of RCC. Furthermore, in clinical trials, the phytochemical coumarin, in conjunction with the histamine 2 receptor blocker cimetidine, was found to exhibit partial and complete remissions in patients with RCC. With comprehensive knowledge of these critical molecular targets and pathways, this information can further inform researchers as to which targets affect oxidative stress, inflammation, proliferation, cell cycle arrest, cell death, angiogenesis, and metastasis. Identifying these targets is crucial, considering that targeted therapy may provide improved outcomes in the treatment of RCC.

The compilation of these various phytochemical mechanisms will further allow researchers to target RCC therapy by accounting for cell line heterogeneity and modulating tumor microenvironments. Marquardt et. al. [290] studied the different histopathological groups of RCC cells lines and found that a substantial number of samples diverge from the RCC RNAseq data. These subgroups were found to be characterized by strengthened mitochondrial and weakened angiogenesis gene signatures. This indicates the importance of identifying the genetically divergent RCC subgroups and developing specific drug therapies for these various genotypes. Multiple potential drug targets for RCC have been investigated for tailored treatment in a study by Argentiero et al. [291]. With the knowledge of subgroups of RCC cell lines with differing histopathological and genetic composition and thus unique tumor microenvironments, the combination therapy for anti-angiogenic and immunomodulatory targets is promising. Therefore, identifying a multitude of anticancer drug targets of phytochemicals can potentially provide a more individualized approach to RCC treatment.

While the results of the in vitro, in vivo, and clinical studies are promising, there are several limitations. Many of the preclinical studies had contradictory results, while some studies could not confirm the mechanisms of action of the phytochemicals. Moreover, many of the phytochemicals were limited to one type of study (in vitro, in vivo, or clinical). To overcome these limitations, additional in vitro, in vivo, and clinical studies should be conducted. Synergistic or additive effects of multiple phytochemicals should be examined to explore additional therapeutic options and evaluate the anticancer capabilities of phytochemicals used together. The addition of more studies to the literature will aid in quantifying the effects of phytochemicals that had contradictory results and may be able to identify new phytochemicals with anticancer effects in RCC.

Another key limitation was that the results for the phytochemicals in RCC consisted of only 15 clinical studies. Many of the clinical trials described were also not randomized, controlled studies, but pooled analyses of prospective studies, which introduced opportunity for bias. Methodological flaws were apparent in the clinical trials, including lack of placebo or control group and small sample sizes. Further clinical trials using randomized, controlled studies should be conducted using the phytochemicals alone and in combinations with others to evaluate their effects on RCC. Additionally, the use of phytochemicals in combination with Food and Drug Administration (FDA)-approved drugs was limited to coumarin and cimetidine. Combination treatment using phytochemicals with FDA-approved chemotherapeutic drugs should be investigated clinically to identify alternative avenues for the treatment of RCC or potential side effects. If the phytochemicals are to make it to clinical application, in vivo studies with animals and larger doses, as well as clinical studies with human subjects, are essential.

A major limitation to the use of phytochemicals is their low bioavailability due to their quick metabolism in the human body. This may lead to poor uptake, deficient targeting, and unacceptable toxicity. This may also limit the therapeutic potential of phytocompounds for the treatment of renal cancer. There has been an increasing interest in developing novel drug delivery tools for improving the pharmacokinetic profile, cellular uptake, and efficacy of phytochemicals, while minimizing the risk for toxic manifestations [292]. Emerging evidence highlights the ability to conjugate or encapsulate the phytochemicals with nanocarriers to enhance their aqueous solubility and bioavailability as well as to overcome multidrug resistance and non-specificity [293,294,295,296]. The improved formulations may expedite the development of phytochemical-based novel anti-renal cancer drugs.

In conclusion, numerous phytochemicals showed promising results for use in the prevention and treatment of RCC. The addition of further studies and establishment of the safety of phytochemical-based cancer drugs may open the door to better treatment in RCC patients. It is our hope that the information provided in this work will be useful for those exploring therapeutic avenues of RCC by creating new treatment protocols with minimal side effects. Overall, based on the available evidence, various phytochemicals exhibit impressive and encouraging potential to meet the ever-growing need to prevent and treat renal cancer.

## Figures and Tables

**Figure 1 cancers-14-03278-f001:**
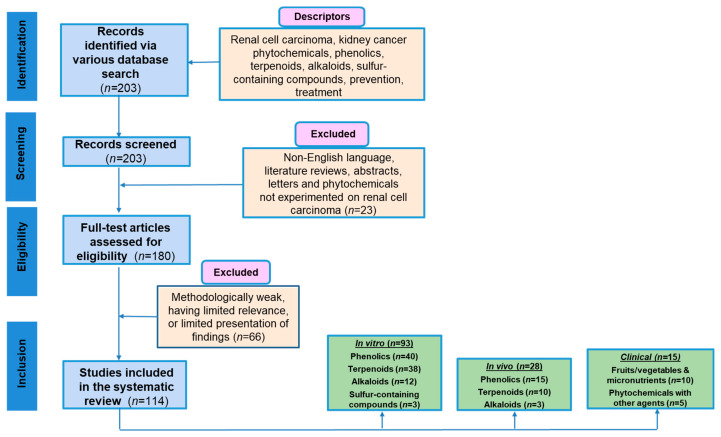
A PRISMA flowchart summarizing the process of literature search and study selection. The total number of in vitro, in vivo, and clinical studies (136) was greater than the number of individual research papers included in our work (114), since many publications contain more than one study type (in vitro or in vivo).

**Figure 2 cancers-14-03278-f002:**
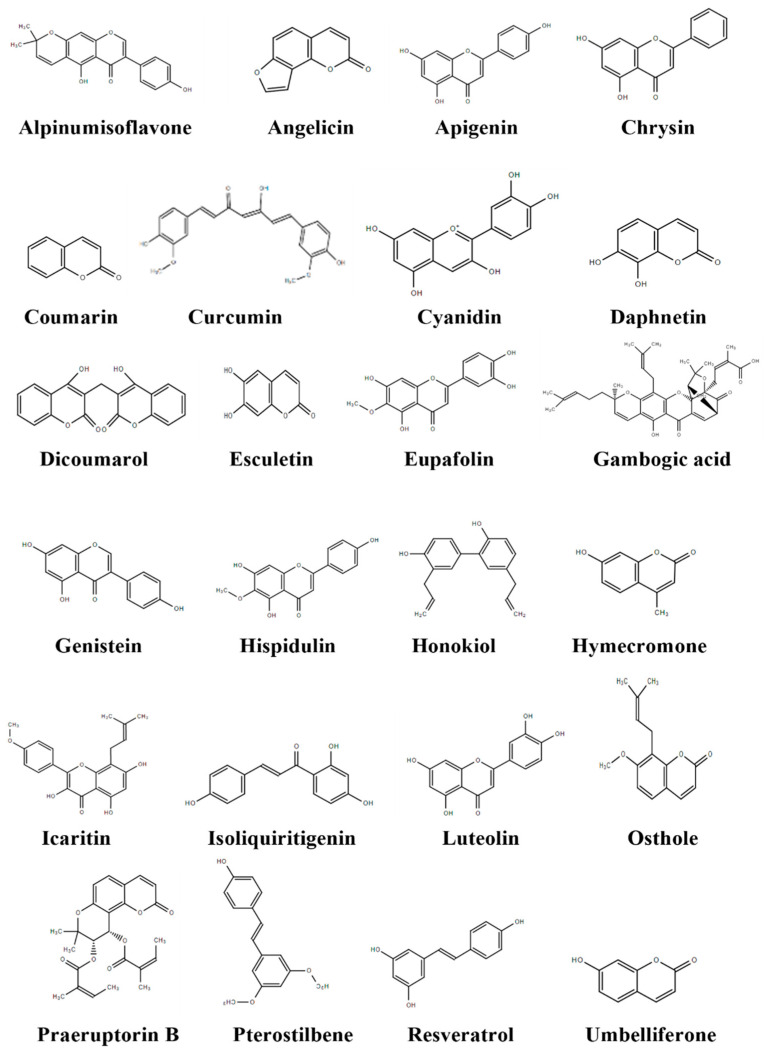
Chemical structures of phenolic compounds that exhibited anticancer effects in RCC. Chemical structures were created and converted using ChemSpider.com (accessed on 14 May 2022).

**Figure 3 cancers-14-03278-f003:**
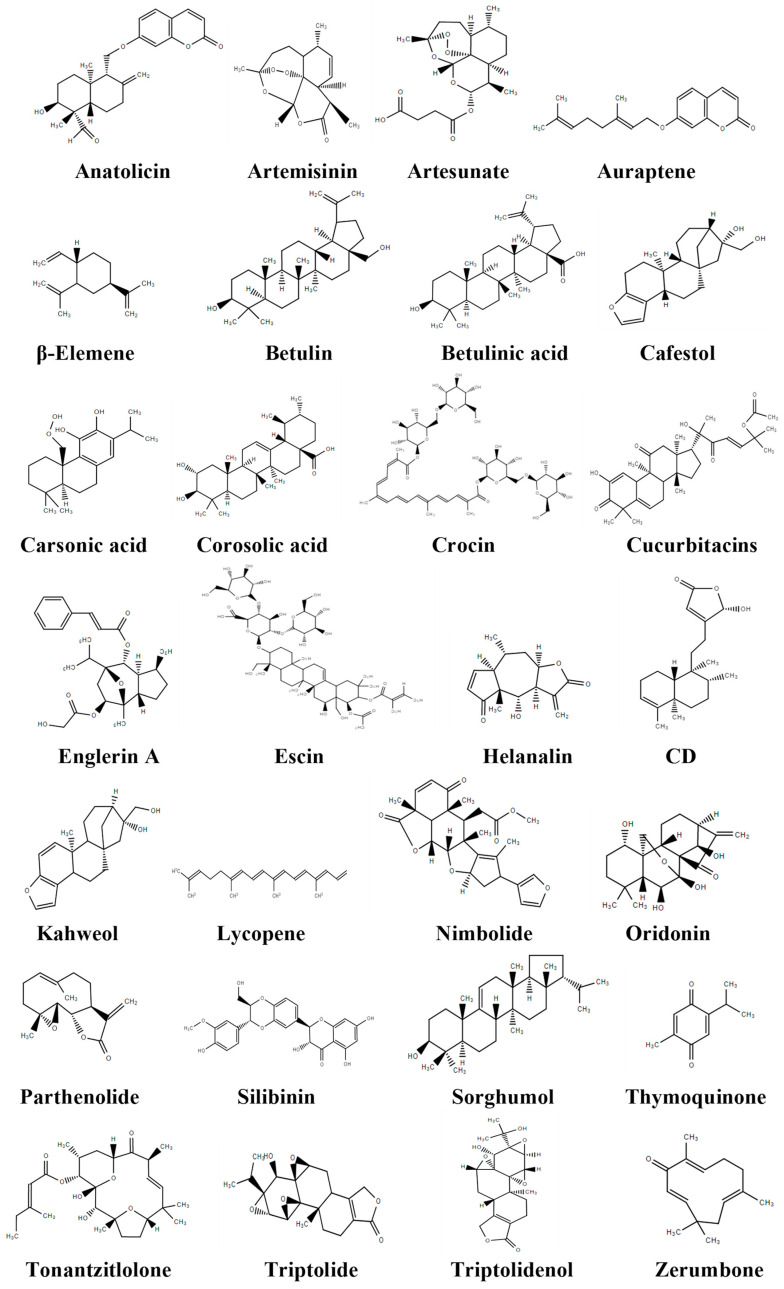
Chemical structures of terpenoids that exhibited anticancer effects in RCC. Chemical structures were created and converted using ChemSpider.com (accessed on 14 May 2022).

**Figure 4 cancers-14-03278-f004:**
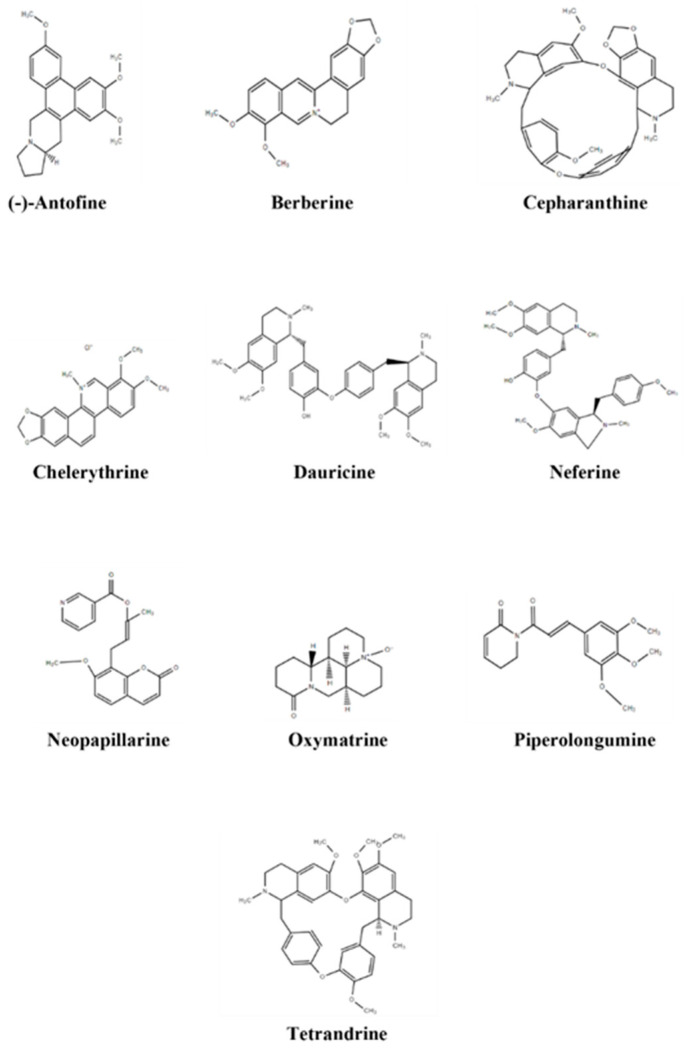
Chemical structures of alkaloids that exhibited anticancer effects in RCC. Chemical structures were created and converted using ChemSpider.com (accessed on 14 May 2022).

**Figure 5 cancers-14-03278-f005:**
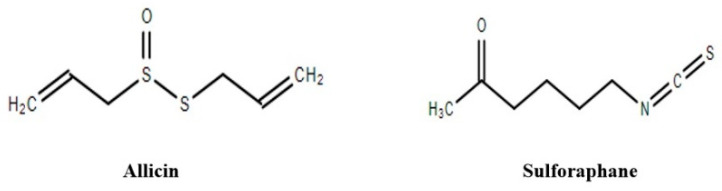
Chemical structures of sulfur-containing compounds that exhibited anticancer effects in RCC. Chemical structures were created and converted using ChemSpider.com (accessed on 14 May 2022).

**Figure 6 cancers-14-03278-f006:**
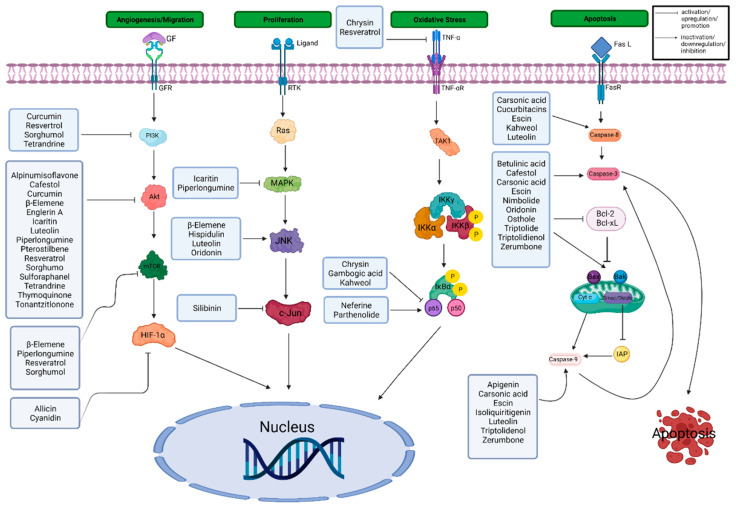
A variety of phytochemicals modulate numerous cellular pathways that are involved in oxidative stress, proliferation, apoptosis, angiogenesis, and migration implicated in renal cancer. These compounds have demonstrated the ability to alter the expression of several key components of these pathways by causing inhibition/downregulation or activation/upregulation. This figure was created using resources available at BioRender.com (accessed on 11 May 2022).

**Table 1 cancers-14-03278-t001:** Potential anti-renal cancer effects and mechanisms of action of phytochemicals based on in vitro studies.

Phytochemicals	Cell Lines Used	Concentrations (Duration)	Anticancer Effects	Mechanisms	References
*Phenolics*					
Alpinumisoflavone	786-O (renal cell carcinoma); Caki-1 (clear cell renal cell carcinoma); RCC4 (clear cell renal cell carcinoma)	2.5–10 μM (48 h)	Suppressed cell growth and invasion	↑Apoptosis; ↑miR-101; ↓RLIP76; ↓p-Akt/Akt	Wang et al., 2017 [39]
Angelicin	Caki (renal cell carcinoma)	50–100 μM (24 h)	No effects alone	Not reported	Min et al., 2018 [40]
Apigenin	ACHN (papillary renal cell carcinoma); 786-O (renal cell carcinoma); Caki-1 (clear cell renal cell carcinoma)	15.4–50.9 μM (24, 48 h)	Inhibited cell proliferation	↑Apoptosis; ↑p53; ↑Bax; ↑caspase-3; ↑caspase-9; ↑DNA damage; ↑γH2AX; ↑G2/M phase arrest	Meng et al., 2017 [41]
Coumarin	786-O (renal cell carcinoma); A-498 (renal cell carcinoma)	10–500 μg/mL (5 days)	Inhibited cell proliferation	Not reported	Myers et al., 1994 [42]
Curcumin	Caki (renal cell carcinoma)	25–100 μM (24 h)	Showed cytotoxic activity	↑Apoptosis; ↓pro-caspase 3; ↑DEVD-pNA cleavage; ↓cIAP1; ↓XIAP; ↓Bcl-2; ↓IAP; ↑cyt c; ↓p-Akt	Woo et al., 2003 [43]
Curcumin	Caki (renal cell carcinoma)	10–30 μM (24 h)	Enhanced therapeutic effects of TRAIL	↑Apoptosis; ↑sub-G1; ↑DEVDase; ↑DR4; ↑DR5; ↑ROS	Jung et al., 2005 [44]
Curcumin	Caki (renal cell carcinoma); ACHN (papillary renal cell carcinoma); A-498 (renal cell carcinoma)	10–30 μM (48 h)	No effects alone	Not reported	Seo et al., 2014 [45]
Curcumin	RCC-949 (renal cell carcinoma)	10–100 μM (24–96 h)	Inhibited proliferation	↑Apoptosis; ↓Bcl-2; ↑Bax; ↑G2/M cell cycle arrest; ↓cyclin B1; ↓PI3K/Akt	Zhang et al., 2015 [46]
Curcumin	Caki-1 (clear cell renal cell carcinoma); OS-RC-2 (renal cell carcinoma)	10–15 μM (48 h)	Decreased proliferation	↑Apoptosis; ↑YAP	Xu et al., 2016 [47]
Cyanidin	786-O (renal cell carcinoma); ACHN (papillary renal cell carcinoma)	0.5–100 μM (24–72 h)	Inhibited growth and migration	↑Apoptosis, ↑G1/M phase arrest; ↑ERG1; ↓SEPW1; ↓HIF2A; ↓E-cadherin; ↓Bcl-2; ↓caspase-3; ↓ROS; ↓autophagy	Liu et al., 2018 [48]
Daphnetin	A-498 (renal cell carcinoma)	5–500 μM (48–96 h)	Inhibited proliferation	↓G1/S; ↑S phase; ↓p-ERK1/2; ↑p38	Finn et al., 2004 [49]
Dicoumarol	ACHN (papillary renal cell carcinoma); A-498 (renal cell carcinoma); Caki (renal cell carcinoma)	25–50 μM (24 h)	Decreased cell viability	↑Apoptosis; ↓Bcl-2; ↓Mcl-1; ↓c-FLIP; ↓NF-κB; ↓CRE; ↑REDD; ↑ATF4; ↑CHOP; ↑GRP78	Park et al., 2014 [50]
Esculetin	786-O (renal cell carcinoma); SN12-PM6 (renal cell carcinoma)	12.5–800 µg/mL (24, 48 h)	Reduced cell viability and inhibited invasion and migration	↑Apoptosis; ↑G0/G1 arrest; ↑G2/M arrest; ↓S phase; ↓cyclin D1; ↓CDK4; ↓CDK6; ↓c-Myc; ↑E-cadherin; ↓N-cadherin; ↓vimentin	Duan et al., 2020 [51]
Eupafolin	Caki (renal cell carcinoma)	20–30 μM (24 h)	No effect alone	None	Han et al., 2016 [52]
Gambogic acid	Caki-1 (clear cell renal cell carcinoma); 786-O (renal cell carcinoma)	0.5–4 μM (48 h)	Enhanced therapeutic effects of sunitinib	↑Sub-G1 population; ↑P21; ↓Bcl-2; ↓VEGF	Jiang et al., 2012 [53]
Genistein	SMKT-R1 (renal cell carcinoma); SMKT-R2 (clear cell renal cell carcinoma); SMKT-R3 (clear cell renal cell carcinoma); SMKT-R4 (renal cell carcinoma)	12.5–100 μg/mL (48 h)	Inhibited proliferation	↑Apoptosis	Sasamura et al., 2004 [54]
Genistein	A-498 (renal cell carcinoma); ACHN (papillary renal cell carcinoma)	10–50 μmol/L (24–120 h)	Induced antiproliferative effects and reduced cell viability	↑G2/M phase arrest; ↑BTG3; ↓DNMTase; ↓DNMT1; ↓DNMT 3b; ↓HDAC	Majid et al., 2009 [55]
Genistein	A-498 (renal cell carcinoma); 786-O (renal cell carcinoma); Caki-2 (papillary renal cell carcinoma)	25 μM (24–72 h)	Inhibited proliferation and invasion	↑Apoptosis; ↓miR-1260b; ↓TCF reporter activity; ↓sFRP1; ↓Dkk2; ↓Smad4	Hirata et al., 2013 [56]
Hispidulin	Caki-1 (clear cell renal cell carcinoma); 786-O (renal cell carcinoma)	12.5–50 μM (48 h)	Inhibited proliferation	↑G0-G1 phase arrest; ↓G2 phase; ↑caspase-3; ↓p-STAT3; ↓Bcl-2; ↓survivin	Gao et al., 2015 [57]
Hispidulin	ACHN (papillary renal cell carcinoma); Caki-2 (clear cell renal cell carcinoma)	10–20 μmol/L (24–72 h)	Decreased cell viability	↑Apoptosis; ↑Fas/Fas ligand; ↑DR5; ↑ceramide; ↓SphK1; ↑ROS/JNK	Gao et al., 2017 [58]
Honokiol	A-498 (renal cell carcinoma)	2.5–80 μmol (72 h)	Suppressed proliferation and migration, inhibited invasion	↑E-cadherin; ↓fibronectin; ↓vimentin; ↓SP cells; ↑miR-141; ↓ZEB2;	Li et al., 2014 [59]
Honokiol	786-O (renal cell carcinoma); A-498 (renal cell carcinoma)	5–80 μM (24 h)	Inhibited migration, suppressed proliferation	↑RhoA/ROCK/MLC	Cheng et al., 2016 [60]
Hymecromone	Caki-1 (clear cell renal cell carcinoma); ACHN (papillary renal cell carcinoma); 786-O (renal cell carcinoma); A-498 (renal cell carcinoma)	16 μg/mL (18, 48 h)	Enhanced therapeutic effects of sorafenib	↑Apoptosis; ↓CD44; ↓RHAMM; ↑caspase-3; ↑caspase-8; ↑caspase-9; ↑cleaved PARP; ↓Mcl-1; ↓p-EGFR; ↓p-Met	Benitez et al., 2013 [61]
Icaritin	786-O (renal cell carcinoma)	1–10 μM (24, 48 h)	Inhibited proliferation	↓Cyclin E; ↓cyclin D1; ↓survivin; ↑caspase-3; ↑cleaved PAR; ↓Bcl-xL; ↓Mcl-1; ↓activated STAT3; ↓JAK2; ↓IL-6-induced JAK2/STAT3; ↓IL-6 induced p-Akt and p-MAPK	Li et al., 2013 [62]
Isoliquiritigenin	Caki (renal cell carcinoma)	5–50 μM (24–72 h)	Decreased cell viability	↑Apoptosis; ↑caspase-3; ↑caspase-7; ↑caspase-9; ↑cleaved PARP; ↑Bax; ↓Bcl-2; ↓Bcl-xL; ↑cyt. c; ↑p53; ↓Mdm2; ↑ROS; ↓STAT3; ↓cyclin D1; ↓cyclin D2; ↑JAK2	Kim et al., 2017 [63]
Luteolin	ACHN (papillary renal cell carcinoma); 786-O (renal cell carcinoma); A-498 (renal cell carcinoma)	10–40 μM (24, 48 h)	Reduced cell viability	↑Apoptosis; ↓p-Akt; ↑p-P38; ↑Ask1; ↑p-JNK; ↑p-ERK; ↑caspase-3; ↑PARP-1; ↓HSP90	Ou et al., 2013 [64]
Luteolin	ACHN (papillary renal cell carcinoma); 786-O (renal cell carcinoma); A-498 (renal cell carcinoma)	10–40 μM (48 h)	Reduced cell viability	↑Apoptosis; ↑caspase-3; ↑caspase-8; ↑caspase-9; ↓Mcl-1; ↓FLIP; ↓p-Akt; ↓p-STAT3,	Ou et al., 2014 [65]
Osthole	786-O (renal cell carcinoma); ACHN (papillary renal cell carcinoma)	20–240 μM (24, 48 h)	Suppressed proliferation, inhibited migration and invasion	↑Apoptosis; ↑caspase-3; ↑Bax; ↓Bcl-2; ↓survivin; ↓MMP2; ↓MMP9; ↑E-cadherin; ↑Beta-catenin; ↓N-cadherin; ↓Smad-3; ↓Snail-1; ↓Twist-1	Liu et al., 2017 [66]
Osthole	Caki (renal cell carcinoma)	20–30 μM (24 h)	Decreased cell viability	↑Apoptosis; ↑sub-G1 population; ↑nuclear condensation; ↑DNA fragmentation; ↑caspase-3; ↓c-FLIP; ↓MMP; ↑cyt. c	Min et al., 2017 [67]
Praeruptorin B	786-O (renal cell carcinoma); ACHN (papillary renal cell carcinoma)	10–50 μM (24 h)	Inhibited cell migration and invasion	↓CTSC; ↓CTSV; ↓p-ERK; ↓p-EGFR; ↓p-MEK	Lin et al., 2020 [68]
Pterostilbene	A-498 (renal cell carcinoma); ACHN (papillary renal cell carcinoma)	5–100 μM (24–72 h)	Displayed antiproliferative effects	↑Apoptosis; ↑Bad; ↑Bax; ↑cyt c; ↑cleaved-caspase 3; ↑cleaved-caspase 9; ↑cleaved-PARP; ↓Bcl-2; ↑S phase cells; ↓p-Akt; ↓p-ERK1/2; ↑γH2AX; ↓Rad1; ↓PCNA	Zhao et al., 2020 [69]
Resveratrol	RCC5430 (renal cell carcinoma)	25–50 μM (12–72 h)	Inhibited cell growth and induced cell death	↑VDR; ↑TRAF1; ↑BGPa; ↑GADD45; ↓GFRA2; ↓HCK; ↓CYP1B1; ↓MUC1; ↓CXCR4	Shi et al., 2004 [70]
Resveratrol	786-O (renal cell carcinoma)	10–40 μM (24–72 h)	Inhibited proliferation	↓VEGF	Yang et al., 2011 [71]
Resveratrol	ACHN (papillary renal cell carcinoma)	7.8125–62.5 µg/mL (12–48 h)	Reduced cell viability, suppressed migration and invasion	↓MMP2; ↓MMP9; ↑acH3K9; ↑acH3K14; ↑acH4K12; ↑acH4K5; ↑acH4K16	Dai et al., 2020 [72]
Resveratrol	Ketr-3 (renal cell carcinoma)	12.5–100 μM (12–72 h)	Suppressed migration and viability	↑Bax; ↓Bcl-2; ↑p53; ↑p-AMPK; ↓p-mTOR; ↑LC3; ↑ATG5; ↑ATG7	Liu et al., 2018 [73]
Resveratrol	ACHN (papillary renal cell carcinoma); A-498 (renal cell carcinoma)	10–200 μM (24–72 h)	Inhibited proliferation, migration and invasion	↑E-cadherin; ↑TIMP-1; ↓N-cadherin; ↓vimentin; ↓snail; ↓MMP2; ↓MMP9; ↓p-Akt; ↓p-ERK1/2	Zhao et al., 2018 [74]
Resveratrol	ACHN (papillary renal cell carcinoma); 786-O (renal cell carcinoma)	25–100 μM (12, 24 h)	Inhibited cell growth and viability	↑Apoptosis; ↑Bax/Bcl-2 ratio; ↓NLRP3; ↓caspase-1; ↑caspase-3; ↓IL-6; ↑E-cadherin	Tian et al., 2020 [75]
Resveratrol	Caki-1 (clear cell renal cell carcinoma)	10–50 μM (24 h)	Enhanced chemosensitivity to paclitaxel and inhibited cell growth	↑Apoptosis; ↓survivin; ↓p-PI3K; ↓p-Akt	Ying-jie et al., 2019 [76]
Resveratrol	Caki-1 (clear cell renal cell carcinoma); 786-O (renal cell carcinoma)	10–50 μM (6 h)	Inhibited proliferation	↑Apoptosis; ↓STAT3; ↓STAT5; ↓JAK1; ↓JAK2; ↓Src kinase; ↑PTPε; ↑SHP-2; ↑S phase; ↓G1/G0; ↑p21; ↓cyclin D1; ↓cyclin E; ↑Bax; ↑p53	Kim et al., 2016 [77]
Umbelliferone	ACHN (papillary renal cell carcinoma); 786-O (renal cell carcinoma); OS-RC-2 (renal cell carcinoma)	5–150 µmol/L (48, 72 h)	Decreased cell proliferation	↑Apoptosis; ↑G1 cell cycle arrest; ↑Bax; ↓Ki-67; ↓MCM2; ↓Bcl-2; ↓CDK2; ↓Cyclin E1; ↓CDK4; ↓CyclinD1; ↓p110γ	Wang et al., 2019 [78]
*Terpenoids*					
Anatolicin	UO31 (RCC); A498 (RCC)	0.017, 0.024 μM (IC_50_) (48 h)	Decreased cell viability	Not reported	Tosun et al., 2019 [79]
Artemisinin	UMRC-2 (clear cell renal cell carcinoma); CAKI-2 (clear cell renal cell carcinoma)	25 μM (24 h)	Inhibited cell growth, inhibited cell invasion and migration	↓c-Myc; ↓ cyclin D1; ↓PCNA, ↓N-cadherin; ↓Vimentin; ↓Snail; ↑E-cadherin	Yu et al. 2019 [80]
Artesunate	Caki (clear cell renal cell carcinoma); ACHN (papillary renal cell carcinoma); A-498 (renal cell carcinoma)	20–50 μM (12–24 h)	Reduced cell viability	↑ROS; ↑RIP1-dependent cell death	Chauhan et al., 2017 [81]
Auraptene	RCC4 (clear cell renal cell carcinoma)	25–100 μM (24 h)	Reduced motility and inhibited metabolism	↓Mitochondrial complex 1; ↓GLUT1; ↓HK2; ↓PFK; ↓LDHA	Jang et al., 2015 [82]
β-elemene	786-O (renal cell carcinoma)	50, 100, 150, 200 μg/mL (24–72 h)	Inhibited cell viability	↑PARP cleavage; ↓Bcl-2, ↓Survivin, ↑p-ERK; ↑p-JNK, ↓PI3K/Akt/mTOR	Zhan et al. 2012 [83]
Betulin	786-O (renal cell carcinoma); Caki-2 (clear cell renal cell carcinoma)	0.5–5 μM (48 h)	Inhibited cell proliferation and reduced cell viability	↑mTOR activation; ↓aerobic glycolysis; ↓p-S6; ↓p-4EBP1; ↓PKM2; ↓HK2	Cheng et al., 2017 [84]
Betulinic acid	786-O (renal cell carcinoma); ACHN (papillary renal cell carcinoma)	5, 10, 20 μg/mL (24–72 h)	Repressed migration and invasion of cancer cells, exerted cytotoxic and cytostatic effects	↑ROS; ↑ MMP2; ↑MMP9; ↑vimentin; ↓TIMP2; ↓E-cadherin; ↑Bax; ↑cleaved caspase-3; ↓Bcl2	Yang et al., 2018 [85]
Cafestol	Caki (clear cell renal cell carcinoma)	10–40 μM (24 h)	Inhibited cell growth and decreased viability	↑Apoptosis; ↑caspase-2; ↑caspase-3; ↑Bim; ↑Bax; ↓cFLIP; ↓Bcl-2; ↓Mcl-1; ↓Bcl-xL; ↑cyt. c; ↓MMP; ↑sub-G1 phase cells; ↓p-Akt; ↓p-STAT3	Choi et al., 2011 [86]
Cafestol; Cafesol+ABT-737	Caki (clear cell renal cell carcinoma)	30 μM (24 h)	Induced cell death	↓Mcl-1; ↑Bim; ↑proteasome activity; ↑sensitivity to ABT-737-mediated apoptosis	Woo et al., 2014 [87]
Carsonic acid	Caki (clear cell renal cell carcinoma); ACHN (papillary renal cell carcinoma); A-498 (renal cell carcinoma)	10–20 μM (24 h)	Inhibited cell viability	↑Apoptosis; ↑sub-G1 population; ↑cleaved-PARP; ↑caspase-3; ↑caspase-8; ↑caspase-9; ↓MMP; ↑cyt c; ↑Bax; ↓c-FLIP; ↑Bcl-2; ↑DR5; ↑Bim; ↑PUMA; ↑ATF4; ↑CHOP; ↑Ca^2+^	Jung et al., 2015 [88]
Carsonic acid	Caki (clear cell renal cell carcinoma)	5–100 μM (24–72 h)	Decreased cell viability	↑Caspase-9; ↑ caspase-7; ↑caspase-3; ↑PARP; ↓Bcl-2; ↓Bcl-xL; ↑Bax; ↑p53; ↓Mdm2; ↑ p27; ↓STAT DNA-binding activity; ↓phosphorylation Y705 and S727; ↓c-Myc; ↓survivin; ↓D-series of cyclins; ↓Src; ↑ROS	Park et al., 2016 [89]
Corosolic acid	Caki (clear cell renal cell carcinoma)	2.5, 5, 10 μM (24 h)	Induces cell death	↓a-tocopherol; ↑lipid peroxidation-dependent non-apoptotic cell death; ↓a-tocopherol; ↓ferrostatin-1; ↓DFO	Woo et al., 2018 [90]
Crocin	A498 (renal cell carcinoma); A704 (renal cell carcinoma)	2.5–10 μM (24–72 h)	Suppressed cell proliferation and migration	↑miR-577 inhibitors; ↓NFIB-wt	Niu et al., 2020 [91]
Cucurbitacins	Caki-1 (clear cell renal cell carcinoma); ACHN (papillary renal cell carcinoma)	1–10 μM (1, 4 h)	Sensitized cells to programmed cell death	↑Apoptosis; ↑caspase-8; ↑TRAIL sensitizing effect	Henrich et al., 2012 [92]
Englerin A	786-O (hypertriploid RCC); A498 (RCC); ACHN (papillary renal cell carcinoma); CAKI-1 (ccRCC); RXF-393 (RCC); SN12C (RCC); TK-10 (ccRCC); UO-31 (RCC)	1–87 nM	Decreased cell growth	Not reported	Ratnayake et al., 2009 [93]
Englerin A	A-498 (renal cell carcinoma); UO-31 (renal cell carcinoma)	1 μM (48 h)	Reduced cell viability	↑ROS; ↑Ca^2+^	Sulzmaier et al., 2012 [94]
Englerin A	A498 (renal cell carcinoma)	50, 100 nm (24, 48 h)	Increased cytotoxicity, blocks cell cycle	↓Akt; ↓ERK; ↓G2/M transition; ↓NEAA; ↑cathepsins; ↑calpains; ↓PI3/Akt; ↑PKCθ; ↑HFS1	Williams et al., 2013 [95]
Englerin A	A-498 (renal cell carcinoma)	0.1–5000 nM 24 h	Inhibited cell proliferation	↑intracellular Ca^2+^; ↑activation TRPC4/C5; ↓TRPA1; ↓TRPV3; ↓TRPV4; ↓TRPM8; ↑PXR	Carson et al., 2015 [96]
Englerin A	cc-RCC (clear cell renal cell carcinoma); A498 (renal cell carcinoma); UO-31 (renal cell carcinoma)	100 nm (24–48 h)	Altered lipid metabolism, inducing ER stress and acute inflammatory response	↑PLIN2; ↑autophagy; ↑PKCθ; ↑ IRF systolic pattern recognition receptors; ↑RIG-I/MDA5; ↑INF-α/ß	Batova et al., 2017 [97]
Escin	786-O (renal cell carcinoma); Caki-1 (clear cell renal cell carcinoma)	6, 12.5, 25, 50, 100 μM (24–72 h)	Exhibited cytotoxic effects, induced cell cycle arrest, ROS production	↓cdc-2; ↑caspase-8; ↑ caspase-9; ↑caspase-3; ↑PARP; ↓ΔΨm; ↓Bcl-2; ↑Bax; ↓IAP proteins like ZIAP and survivin; ↑ ROS; ↓Fas death receptor; Fas-L; FADD	Yuan et al., 2017 [98]
Helenalin	Caki (clear cell renal cell carcinoma); ACHN (papillary renal cell carcinoma)	2–6 μM (24 h)	Decreased cell viability	↑Apoptosis; ↑sub-G1 population; ↑caspase-3; ↑ROS, ↑PARP; ↑ATF4; ↑REDD1; ↑CHOP	Jang et al., 2013 [99]
16-hydroxyclerod-3,113-dien-15,16-olide	ccRCC (clear cell renal cell carcinoma); A-498 (renal cell carcinoma); 786-O (renal cell carcinoma)	10–40 μM (24–72 h)	Inhibited cell proliferation and decreased cell viability	↑ROS; ↓MMP; ↓mitochondrial cyt. c; ↑cytosolic cyt. c, ↓ Bcl-2; ↓hsp70; ↑caspase-3; ↑cleaved PARP; ↓p-Akt; ↓mTOR; ↓MEK1/2; ↓ERK1/2; ↓c-Myc; ↓HIF-2a; ↑FOXO2a	Liu et al., 2017 [100]
Kahweol	Caki (clear cell renal cell carcinoma)	5–20 μM (24 h)	Decreased cell viability	↑Apoptosis; ↓c-FLIP; ↓Bcl-2; ↑PARP cleavage; ↑caspase-8; ↓NF-kB p65	Um et al., 2010 [101]
Kahweol	Caki (clear cell renal cell carcinoma); ACHN (papillary renal cell carcinoma); A498 (renal cell carcinoma)	20 μM (24 h)	Decreased cell viability	↑sub-G1 population; ↑PARP cleavage; ↓c-FLIP; ↓Mcl-1	Min et al., 2017 [102]
Nimbolide	786-O (renal cell carcinoma); A-498 (renal cell carcinoma)	1–6 μM (24 h)	Decreased cell viability	↑Apoptosis; ↑G2/M cell cycle arrest; ↑p-p53; ↑p-cdc2; ↑p-cdc25c; ↓cyclin A; ↓cyclin B; ↓cdc2; ↓cdc25c; ↑DNA damage; ↑γ-H2AX; ↑cleaved-caspase-3; ↑cleaved-PARP; ↓pro-caspase-8; ↓Mcl-1; ↓Bcl-2; ↑Bax; ↑DR5; ↑CHOP	Hsiesh et al., 2015 [103]
Oridonin	786-O (renal cell carcinoma)	10, 20, 40 μM (24 h)	Increased cell cytotoxicity	↑Apoptosis; ↑necroptosis; ↑cleaved caspase-3; ↑Bax; ↓Bcl-2; ↑RIP-1; ↑RIP-3; ↑LDH; ↑HGMB1; ↑Parp-1; ↓GSH; ↑p-JNK; ↑p-ERK; ↑p-P38	Zheng et al., 2018 [104]
Parthenolide	OUR-10 (renal cell carcinoma) ACHN (papillary renal cell carcinoma)	0.1–5 μM (24 h)	Decreased cell growth and proliferation	↑NF-κB p65; ↑IkBα; ↑p-NF-kB	Oka et al., 2007 [105]
Silibinin	786-O (renal cell carcinoma)	10–50 μM (24 h)	Inhibits migration and invasion of cancer cells	↓MMP-9; ↓MMP-2; ↓u-PA; ↑TIMP-2; ↑PAI-1; ↓ERK 1/2 phosphorylation; ↓p38; ↓NF-kB; ↓c-Jun; ↓c-Fos	Chang et al., 2011 [106]
Sorghumol	A-498 (renal cell carcinoma)	10–40 μM (24 h)	Reduced cell proliferation	↑Apoptosis; ↑arrest in G2/M; ↓p-mTOR; ↓p-PI3K; ↓p-Akt	Li et al., 2019 [107]
Thymoquinone	Caki (clear cell renal cell carcinoma); A498 (renal cell carcinoma); ACHN (papillary renal cell carcinoma)	25–75 μM (24 h)	Decreased cell viability	↑PARP; ↓c-FLIP; ↓Bcl-2; ↓NF-κB; ↑ROS; ↑cyt c; ↓MMP	Park et al., 2016 [108]
Thymoquinone	786-O-S13 (RCC)	5–20 µM (24 h)	Decreased cell movement, repressed migration and invasion of cancer cells	↓MMP-2; ↓u-PA; ↓p-phosphatidylinositol 3-kinase; ↓p-Akt; ↓p-Src; ↓p-Paxillin; ↓fibronectin; ↓N-cadherin; ↓Rho A; ↓TGF-β-promoted u-PA activity; ↑adhesion type I collagen; ↑adhesion type IV collagen	Liou et al., 2019 [109]
Tonantzitlolone	ccRCC (clear cell renal cell carcinoma); 786-O (renal cell carcinoma); A498 (renal cell carcinoma)	1–5 µM (24 h)	Increased cell cytotoxicity	↑PKCθ; ↑PKCα; ↑p-IRS1; ↓p-Akt; ↓glucose uptake; ↑PKCθ-dependent HSF1 phosphorylation	Sourbier et al., 2015 [110]
Tonantzitlolone	A498 (renal cell carcinoma); HEK293 (human embryonic kidney cell)	10–1000 nM (24 h)	Increased cell cytotoxicity	↑intracellular Ca^2+^; ↑TRPC5 channels	Rubaiy et al., 2018 [111]
Triptolide	786-O (renal cell carcinoma); OS-RC-2 (renal cell carcinoma)	12.5–200 nM (24, 48 h)	Decreased cell viability	↑Apoptosis; ↑caspase-3; ↑cyt. c; ↑Bax; ↓Bcl-2; ↓Bcl-xL; ↑cell cycle arrest at S phase; ↓Rb; ↓A/CDK1; ↓CDK2; ↓B/CDK; Induced cell cycle arrest	Li et al., 2011 [68]
Triptolide	A498 (renal cell carcinoma); Caki-1 (clear cell renal cell carcinoma); ACHN (papillary renal cell carcinoma); 786-O (renal cell carcinoma); 769-P (renal cell adenocarcinoma)	10–100 nM (24 h)	Enhanced therapeutic effects of TRAIL	↑TRAIL-R2; ↓HSP70; ↑Apoptosis	Brincks et al., 2015 [112]
Triptolidenol	786-O (renal cell carcinoma); Caki-1 (clear cell renal cell carcinoma); ACHN (papillary renal cell carcinoma)	30–500 nM (48 h)	Reduced cell proliferation	↑Apoptosis; ↑Bax; ↓Bcl-2; ↑cyt. c; ↑cleaved-caspase-3; ↑cleaved-caspase-9; ↑cleaved-PARP; ↑S phase arrest; ↓cyclin A2; ↓cyclin D1; ↓cyclin E1; ↓CDK2; ↓CDK4; ↓COX-2; ↓NF-κB; ↓MMP	Jin et al., 2021 [113]
Zerumbone	786-O (renal cell carcinoma); 769-P (renal cell adenocarcinoma)	10–50 µM (24–72 h)	Decreased cell viability	↑caspase-3; ↑caspase-9; ↓Gli-1; ↓Bcl-2	Sun et al., 2013 [114]
Zerumbone	786-O (renal cell carcinoma); Caki-1 (clear cell renal cell carcinoma)	5–50 µM (24 h)	Inhibited colony formation	↓Ki-67; ↓Bcl-2; ↓Bcl-xL; ↑caspase-3; ↑PARP; ↓cyclin D1; ↓SHP-1; ↓Mcl-1; ↓Survivin; ↓MMP-9; ↓VEGF; ↓CD31; ↓constitutive p-STAT3 activation; ↓STAT3-DNA binding activity; ↓phosphorylation of c-Src; ↓p-JAK1; ↓p-JAK2	Shanmugam et al., 2015 [115]
*Alkaloids*					
(-)-Antofine	Caki-1 (ccRCC)	25 nM (24 h)	Decreased cell growth	↓Met; ↓STAT3; ↓Grb2; ↓RAC1; ↓Src; ↓ERK1	Song et al., 2015 [116]
Berberine	Caki (clear cell RCC)	60 µM (24 h)	Increased cell death	↓c-FLIP; ↓Mcl-1	Lee et al., 2011 [117]
Berberine	ACHN (papillary renal cell carcinoma); 786-O (hypertriploid RCC)	5–320 µM (24, 48 h)	Decreased cell viability and increased cell death	↑ROS; ↑caspase 3; ↑PLK3; ↓FIGF; ↓TERT; ↓formate; ↓lactate; ↑lysine	Lopes et al., 2020 [118]
Cepharanthine	Caki (clear cell RCC); ACHN (papillary renal cell carcinoma); A498 (RCC)	10–15 µM (18 h)	Decreased cell viability	↓Survivin; ↓cellular FLICE inhibitory protein (cFLIP); ↓STAMBPL1; ↑USP53; ↑DR5	Shahriyar et al., 2018 [119]
Chelerythrine	Caki (clear cell RCC); 786-O (hypertriploid RCC)	0.3–40 µM/L (24 h)	Anti-tumor activity of CHE against RCC cells and CHE induced cell cycle arrest and cell viability.	↓STAT3; ↑ROS; ↑Cle–PARP; ↓Bcl-2; ↑Bax; ↑G2/M phase; ↓cyclin-dependent kinase 1 (CDC2); ↓cyclin B1; ↓MDM-2; ↑ROS; ↑p-elF2alpha; ↑ATF4; ↓Y705 residues of STAT3	He et al., 2020 [120]
Dauricine	786-O (hypertriploid RCC); A498 (RCC); ACHN (papillary renal cell carcinoma); Caki (clear cell RCC)	5–20 µM (24 h)	Decreased cell growth and decreased cell viability and increased cell death	↓Cyclin D1; ↓CDK4; ↓CDK2; ↓pro-caspase-9; ↓Bcl2; ↓MCL-1; ↑p21; ↑Bax	Zhang et al., 2018 [121]
Neferine	Caki-1 (ccRCC); ACHN (papillary renal cell carcinoma); A498 (RCC)	5–25 µM (24 h)	Decreased cell viability	↑Bcl-2; ↑p65; ↓NF-κB	Kim et al., 2019 [122]
Neopapillarine	A498 (RCC); UO31 (RCC)	20, 67 µM	Decreased cell growth	Not reported	Tosun et al., 2020 [123]
Oxymatrine	A498 (RCC); SW839 (ccRCC)	0.5–10 mg/mL (24–72 h)	Decreased cell viability	↓Ki-67; ↓cyclin D1; ↓CDK6; ↑p27; ↑caspase-3; ↑cleaved PARP; ↓Vimetin; ↓MMP2; ↓MMP9; ↑E-cadherin; ↑GSK-3β; ↓β-catenin translocation	Jin et al., 2020 [124]
Piperlongumine	786-O (RCC)	2.5–20 µM (24, 72 h)	Decreased cell viability and cell growth	↑Apoptosis; ↓LC3-II; ↓p-mTORC1; ↓GSK-3β; ↓TSC2; ↓Akt/mTORC1; ↑ROS; ↓p-Ser757 ULK1	Makhov et al., 2014 [125]
Piperlongumine	RCC 786-O (hypertriploid RCC); PNX0010 (ccRCC)	2.5–10 µM/L (24 h)	Inhibited cell growth and cell viability	↓c-Met; ↓ERK; ↓MAPK; ↓STAT3; ↓NF-κB; ↓Akt; ↓mTOR	Golovine et al., 2015 [126]
Tetrandrine	RCC 786-O (hypertriploid RCC); 769-P (RCC)	0.05–10 mg/mL (24 h)	Inhibited cell growth	↓NF-κB; ↓MMP-9; ↓p-Akt; ↓p-PI3K; ↓p-PDK1	Chen et al., 2017 [127]
*Sulfur-containing compounds*					
Allicin	RCC-9863 (RCC)	0.016, 0.05, 0.1 mg/mL (48 h)	Decreased cell viability; decreased cell differentiation; decreased chemotactic ability	↓Bcl-2; ↓VEGF; ↓HIF-1α; ↑Bax	Song et al., 2015 [128]
Sulforaphane	Caki-1^res^, KTCTL-26^res^, A498^res^, Caki-1^par^, KTCTL-26^par^, and A498^par^	1.25 to 20 µM (24–72 h)	Decreased cell growth and migration	↑G2/M; ↑S phase; ↓Cdk1; ↓pCdk2; ↓Cdk2; ↓cyclin A; ↓cyclin B; ↓p27; ↓p-Akt; ↑p19; ↓p-Rictor; ↓α5; ↓α6; ↓β1; ↓β4; ↓β3; ↑β3; ↓integrin α2	Juengel et al., 2016 [129]
Sulforaphane	A498 (RCC); Caki-1 (ccRCC); KTCL-26 (RCC)	Short term: 5 µM (24 h) Long term: 5 µM (8 wk)	Decreased cell growth	Short term: ↑G2/M; ↑cdk1; ↑cdk2; ↑cyclin A; ↑cyclin B; ↓p-Akt; ↓p-Raptor Long term: ↑G0/G1; ↓cdk1; ↓cdk2; ↓cyclin A; ↓cyclin B; ↓p-Akt; ↓p-Raptor	Juengel et al., 2017 [130]

Symbols: ↑, increased or upregulated; ↓, decreased or downregulated; IC_50_, half-maximal inhibitory concentration.

**Table 2 cancers-14-03278-t002:** Potential anti-renal cancer effects and mechanisms of action of phytochemicals based on in vivo studies.

Phytochemicals	Animal Tumor Models	Anticancer Effects	Mechanisms	Dose (Route)	Duration	References
*Phenolics*						
Alpinumisoflavone	BALB/c male nude mice xenografted with 786-O cells	Suppressed tumor growth and metastasis	↑Apoptosis; ↑miR-101; ↓RLIP76; ↓p-Akt/t-Akt ratio	40, 80 mg/kg (s.c.)	Not reported	Wang et al., 2017 [39]
Apigenin	BALB/c male nude mice xenografted with ACHN cells	Reduced tumor growth and volume	↓Ki-67	30 mg/kg (i.p.)	21 days	Meng et al., 2017 [41]
Chrysin	Male Wistar rats induced by Fe-NTA and DEN	Suppressed renal carcinogenesis	↑GSH; ↑GR; ↑GPx; ↑catalase; ↑SOD; ↓ODC; ↓proliferation; ↓IL-6; ↓TNFα; ↓PGE2; ↓NF-κBp65; ↓COX-2; ↓iNOS	20, 40 mg/kg (p.o.)	16 weeks	Rehman et al., 2013 [131]
Coumarin	BALB/c female mice injected with tumor inducing BMK cells	Suppressed tumor growth and metastasis	Not reported	20 mg/kg (i.p.)	3 weeks	Ruiz-Marcial, 2007 [132]
Cyanidin	BALB/c nude male mice xenografted with ACHN cells	Inhibited tumor growth	↓Ki-67;	6 mg/kg (i.p.)	4 weeks	Liu et al., 2018 [48]
Eupafolin	BALB/c male nude mice xenografted with Caki cells	Suppressed tumor growth	↑Apoptosis	10 mg/kg (i.p.)	3 weeks	Han et al., 2016 [52]
Gambogic acid	BALB/c male mice xenografted with Caki-1 cells	Decreased tumor size	↓Ki-67; ↓CD31; ↓p65	5 mg/kg (p.o.)	4 weeks	Jiang et al., 2012 [53]
Genistein	C57 BL 6 female mice injected with SMKT R-1 cells	Suppressed angiogenesis	Not reported	100 μg/mL (dorsal air sac model)	12 h	Sasamura et al., 2004 [54]
Hispidulin	BALB/c nu/nu male mice xenografted with Caki-1 cells	Decreased tumor size	Not reported	20 mg/kg (i.p.)	4 weeks	Gao et al., 2015 [57]
Hispidulin	BALB/c nu/nu male mice xenografted with Caki-2 cells	Suppressed tumor growth	↑Apoptosis; ↑cleaved caspase-3; ↑ceramide; ↑p-JNK; ↓SphK1	20, 40 mg/kg (i.p.)	27 days	Gao et al., 2017 [58]
Honokiol	BALB/c nude mice xenografted with A-498 cells	Delayed tumor growth	↑E-cadherin; ↓ZEB2; ↓vimentin; ↓fibronectin	20 μmol (s.c.)	21 days	Li et al., 2014 [59]
Icaritin	BALB/c female mice xenografted with RENCA cells	Inhibited tumor growth	↓STAT3; ↓Bcl-xL; ↓cyclin E; ↓VEGF; ↓tumor vasculature	10 mg/kg (peritumoral injection)	7 days	Li et al., 2013 [62]
Resveratrol	BALB/c female mice xenografted with Renca cells	Inhibited tumor growth	↑CD8+ T cells; ↑perforin; ↑granzyme B; ↑FasL; ↑Fas; ↓Tregs; ↑IFN-ℽ; ↓IL-6; ↓IL-10; ↓VEGF	1–5 mg/kg (i.p.)	22 days	Chen et al., 2015 [133]
Resveratrol	Male nude mice xenografted with ACHN cells	Reduced tumor volume	↑Apoptosis; ↑Bax/Bcl-2 ratio; ↓NLRP3	60 mg/kg (gavage injection)	40 days	Tian et al., 2020 [75]
Resveratrol	Male Wistar rats induced with clear cell renal cell carcinoma	Displayed antioxidant and anti-inflammatory properties	↑Nrf2; ↑HO-1; ↑CAT; ↑SOD; ↑GPx; ↓TBARS; ↓xanthine oxidase; ↓LDH; ↓TGF-β1; ↓IL-6; ↓TNF-α; ↓STAT3; ↑caspase-3; ↓NF-κB	30 mg/kg (p.o.)	24 weeks	Kabel et al., 2018 [134]
*Terpenoids*						
Artemisinin	Male nude mice subcutaneously injected with UMRC-2 cells	Reduced tumor growth and volume	↓c-Myc; ↓cyclin D1; ↓PCNA; ↓N-cadherin; ↓Vimetin; ↓Snail; ↑E-cadherin; ↓p-Akt	20 mg/kg (p.o.)	2 weeks	Yu et al., 2019 [80]
Auraptene	Nude male mice xenografted with RCC4 cells	Reduced tumor growth	↓Vegf-a; ↓angiogenesis	100 μM (s.c.)	25 days	Jang et al., 2015 [82]
Betulinic acid	Female BALB/c nude mice injected with 786-O cells	Impedes tumor growth	↓Ki-67-positive; ↓MMP9	5, 10 mg/kg (p.o.)	15 days	Yang et al., 2018 [85]
Cafestol; Cafestol + ABT-737	Male BALB/c-nude mice injected with Mcl-1-overexpressing Caki cells in combination with ABT-737	Suppressed tumor growth and increased cell death	↑Apoptosis; ↑caspase 3	75 mg/kg (i.p.)	2 weeks	Woo et al., 2014 [87]
Lycopene	Female Eker rats that carry the Eker TSC2 mutation	Reduced number and size of renal carcinomas	Not reported	100, 200 mg (p.o.)	18 months	Sahin et al., 2015 [135]
Oridonin	Nude mice injected subcutaneously with 786-O cells	Decrease in tumor weight	↑cleaved caspase-3; ↑Bax, ↓Bcl-2; ↑RIP-1 and RIP-3; ↑LDH; ↑HGMB1; ↑Parp-1; ↓GSH; ↑p-JN; p-ERK; p-P38	20 mg/kg (i.p.)	32 days	Zheng et al., 2018 [104]
Parthenolide	Nude male mice inoculated with OUR-10 cells	Reduction in tumor volume	↓NF-κB; ↓NF-κB p65; ↓p-NF-κB; ↑IκBα; ↓Bcl-xL; ↓COX-2; ↓MMP-9	10 mg/kg (p.o.) 3 μg/mouse (s.c.)	6 weeks	Oka et al., 2007 [105]
Silibinin	Nude mice with subcutaneous inoculation of RCC 786-O cells	Reduction in tumor volume	Not reported	200 mg/kg (p.o.)	44 days	Chang et al., 2011 [106]
Thymoquinone	Five-week-old male C57BL/6 mice	Inhibited RCC spread to lungs	↓MMP-2; ↓u-PA; ↓p-phosphatidylinositol 3-kinase; ↓p-Akt; ↓p-Src; ↓p-Paxillin; ↓fibronectin; ↓N-cadherin; ↓Rho A; ↓TGF-β-promoted u-PA activity; ↑adhesion type I collagen; ↑adhesion type IV collagen	10 mg/kg or 20 mg/kg per day (p.o.)	42 days	Liou et al. 2019 [109]
Zerumbone	Athymic nu/nu female mice inoculated with 786-O cells	Decreased tumor growth	↓Ki-67; ↓Bcl-2; ↑caspase-3; ↓CD31; ↑SHP-1; ↓p-STAT2	50 mg/kg (i.p.)	6 weeks	Shanmugam et al., 2015 [115]
*Alkaloids*						
(-)-Antofine	Athymic mice (BALB/c-nu) injected with A Caki-1 cells	Decreased tumor growth	↓Met-mediated STAT	5 mg/kg, five times a week (p.o.)	42 days	Song et al., 2015 [116]
Oxymatrine	BALB/c nude mice xenografted with A498 or SW839 cells	Suppressed tumor cell growth	↑Apoptosis; ↓β-catenin	50 mg/kg (i.p.)	28 days	Jin et al., 2020 [124]
Piperlongumine	C.B17/Icr-scid mice inoculated with PNX0010 cells	Decreased tumor growth	↓c-Met	Treated with PL or PL-Di (20 mg/kg, i.p., 3 times/week)	18 days	Golovine et al., 2015 [126]

Symbols: ↑, increased or upregulated; ↓, decreased or downregulated.

**Table 3 cancers-14-03278-t003:** Completed clinical studies on dietary agents/phytochemicals for renal cancer prevention and intervention.

Agents Tested	Study Subjects	Study Type	Study Population	No. of Patients/Control Subjects	Intervention	Main Findings/Objectives	References
*Fruits/vegetables and micronutrients*							
Cruciferous and dark-green vegetables (α-carotene, β-carotene, lutein, β-cryptoxanthin)	1276 cases of confirmed RCC among residents of Los Angeles, CA	Case–control	USA	1276/1204	Dietary intake of cruciferous vegetables was determined via a validated food frequency questionnaire	Dietary intake of cruciferous vegetables was associated with a decreased risk that may be explained by a variety of carotenoids.	Yuan et al., 1998 [275]
Carotenoids, vitamin A, and C	88,759 women in the Nurses’ Health Study and 47,828 men in the Health Professionals Follow-up Study with 248 developing RCC	Prospective cohort study	USA	136,587	Dietary intake of fruit and vegetables was determined via a validated semiquantitative food frequency questionnaire implemented every 2–4 years	Dietary intake of fruits and vegetables was associated with a decreased risk of RCC in men, but not in women. Intake of vitamin A and C and carotenoids from food was inversely associated with a decreased risk of RCC in men.	Lee et al., 2006 [276]
Carotenoids including α-carotene and lutein/zeaxanthin	1478 cases of RCC among 530,469 women and 244,483 men	A pooled analysis of 13 prospective studies	USA	774,952	Dietary intake of fruit and vegetables was determined via a validated food frequency questionnaire implemented	Dietary intake of fruits and vegetables was associated with a decreased risk of RCC and may be due to certain carotenoids contained within the diet.	Lee et al., 2009 [277]
Vitamin E and vitamin C	Hospitalized patients with confirmed RCC	Case–control	Italy	767/1534	Dietary habits were determined by food frequency questionnaire	A significant inverse relationship between dietary vitamin E intake and borderline significant inverse relationship of vitamin C intake with associated risk of RCC.	Bosetti et al., 2006 [278]
Flavonoids	Patients with RCC	Case–control	Italy	767/1534	Intake of flavonoids was determined via diet questionnaire	Inverse association between flavonoid intake and the risk of developing RCC.	Bosetti et al., 2007 [279]
Carotenoids, vitamin C, and E	1138 cases of confirmed RCC	Case-control	Canada	1138/5039	Dietary intake was determined via a 69-item food questionnaire	Dietary intake of β-carotene and lutein/zeaxanthin was inversely associated with RCC. This association was more pronounced in women as well as overweight and obese subjects. There was no significant association between vitamin C, vitamin E, β-cryptoxanthin, and lycopene with RCC.	Hu et al., 2009 [280]
β-cryptoxanthin and vegetable fiber	Residents of Iowa between the ages of 40–85 with RCC	Case–control	USA	323/1827	Dietary habits were determined by food frequency questionnaire	Dietary intake of vegetable fiber was associated with a decreased risk of RCC, but this was not observed for fruit or grain fiber. Additionally, β-cryptoxanthin was the only micronutrient associated with a significant decreased risk of RCC.	Brock et al., 2012 [281]
α-carotene, β-carotene, lutein zeaxanthin, lycopene, vitamin A, folate, thiamin, vitamin C, α-tocopherol, β-tocopherol, γ-tocopherol, selenium, and β-cryptoxanthin	European, American, and African American men and women with newly diagnosed RCC	Population-based case–control	USA	1142/1154	Dietary micronutrient intake was derived from an interviewer-administered diet history questionnaire (DHQ) and queried frequency of intake for foods, beverages, multivitamin, vitamin C, and vitamin E supplements, for a total of 80 line items.	Inverse associations with RCC risk were observed for α-carotene, β-carotene, lutein zeaxanthin, lycopene, vitamin A, folate, thiamin, vitamin C, α-tocopherol, β-tocopherol, γ-tocopherol, and selenium. A trend for β-cryptoxanthin was suggested among EA but not AA or the total sample (P-interaction = 0.04).	Bock et al., 2018 [282]
Lycopene	Postmenopausal women	Prospective cohort study	USA	96,196	Dietary micronutrient intake was estimated from the baseline WHI food frequency questionnaire, and data on supplement use were collected using an interview-based inventory procedure. Study conducted over 11–12 years	Inverse relationship with lycopene intake and risk of developing RCC. No associations with intake of α-carotene, β-carotene, β-cryptoxanthin, lutein plus zeaxanthin, vitamin C, and vitamin E.	Ho et al., 2015 [283]
β-carotene, α-tocopherol, and retinol	Male smokers	Prospective cohort study	Finland	27,062	276-item dietary questionnaire with pre-randomization serum β-carotene, retinol, and α-tocopherol levels. Additionally daily supplementation with β-carotene (20 mg) and/or vitamin E (50 mg) during up to 19 years follow-up	Overall, no association between fruit, vegetables, antioxidant nutrients and risk of renal cell carcinoma.	Bertoia et al., 2010 [284]
*Phytochemicals with other agents*							
Coumarin + cimetidine	Patients with metastatic RCC	Pilot study	Not specified	45	100 mg coumarin p.o. daily followed by 300 mg cimetidine four times a day on day 15	Three patients with complete remission, eleven patients with partial remission. Twelve patients experienced stabilization of disease.	Marshall et al., 1987 [285]
Coumarin + cimetidine	Patients with RCC	Phase II trial	Germany	31	100 mg coumarin p.o. q.d. followed by 400 mg cimetidine four times a day on day 15	Two patients achieved partial responses. One mixed response, and five with stable disease.	Hermann et al., 1990 [286]
Coumarin + Cimetidine	Patients with metastatic RCC	Pilot	USA	50	100 mg coumarin p.o daily starting on day 1 and cimetidine 300 mg p.o four times a day starting on day 15. When disease progressed, coumarin was escalated to 100 mg orally four times a day	Three patients achieved a partial response, one of those after dose escalation. In addition, one patient had a minor response, then progressing disease, and again had a minor response after dose escalation.	Dexeus et al., 1990 [287]
Coumarin + cimetidine	Patients with metastatic RCC	Pilot study	Austria	39	400 mg cimetidine p.o. daily followed by 100 mg coumarin p.o. daily after 1 week	Complete remission and partial remission of 5 patients.	Kokron et al., 1991 [288]
Coumarin + IFN-α	Patients with RCC	Randomized prospective multicenter phase III trial	Austria	148	100 mg coumarin p.o. daily, IFN-a-2b (5 MU) 5 × weekly s.c.	No significant increase in response rate or prolongation of survival times.	Sagaster et al., 1995 [289]

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
