# Peer review of "Phytochemicals for the Prevention and Treatment of Renal Cell Carcinoma: Preclinical and Clinical Evidence and Molecular Mechanisms"

_cancers, 2022, doi:10.3390/cancers14133278_

Round 1

Reviewer 1 Report

In their review Essa M. Bajalia et al. analyzed potential role of phytochemicals in RCC treatment. They described effects of more than 60 phytochemicals in both in vitro and in vivo studies. The study addressed a potentially interesting issue. Renal cell carcinoma (especially metastatic form) is therapy resistant. Knowledge about effects of natural compounds on RCC could help to develop new therapeutics. I recommend the study of Essa M. Bajalia et al. for publication in the Cancer Biomarkers after minor changes.

1)      Authors often write about Caki cell line ex. Verses 245, 301, 325 etc., however there are two RCC cell lines which could be named Caki - Caki-1 and Caki-2 (Caki-1 is cell line derived from skin metastasis of ccRCC while Caki-2 is cell line derived from primary renal cell carcinoma). Authors should specify which cell line was tested in discussed articles.

2)      In descriptions of some phytochemicals authors give information about the table in which experiments are summarized (e.g. 3.3.1 verse 1058, 1066) in others not (e.g. 3.3.8). It should be standardized - authors should add information about tables to all paragraphs or remove it.

3)      Detailed list of small mistakes which must be corrected before publication”

3.1.16 Hymecrone

 - verse 436  “Hymecrone (…) produced a downregulation” it should be changed

 -  Sentence from verse 459 starting from Confirmation of apoptosis - should be corrected - it is hard to follow

3.1.202 - verse 494 - chemical name of osthole as Figure 2 are in square brackets([]) instead of normal brackets (())

3.1.24 verses 596 and 597 p110g is not PI3K member protein - p110g is an isoform of catalytic subunit of PI3K kinase.

3.1.24 verse 596  - it should be PI3K instead P13K.

3.2.3 verse 638 there is information about two cell lines while in the Table1 3 cell lines are listed

3.2.4  verses 647-648 and from 656 the same results of Jang et al. are described

3.2.9 verse 748 MMP is Mitochondrial Membrane Potential (there is no such abbreviation in abbreviations list it must be corrected in text and in abbreviations list)

3.2.11 Information about crocin effects on miR-577 expression is doubled. (verses 777and 778) the sentence form verses 778-779 should be corrected - it does not sound scientific.

3.3.2 sentence in verses 1082- 1083 should be corrected

3.3.9 verse 1174 c-Met is not mesenchymal-epithelial factor (c-Met is HGF receptor) it should be corrected in text and in abbreviation list

3.4.2 verse 1208 - The sentence from verse 1208 “Everolimus (…. ) can cause resistance (…)” must be corrected. Everolimus is a drug and during therapy patients could develop resistance for this drug; verse 1223 sulforaphane was used to prevent cells from development of resistance not to overcome it.

Abbreviations

c-Met is not mesenchymal-epithelial factor

MMP are matrix metaloproteinases but in text (verse 748) MMP is also Mitochondrial Membrane Potential

SRC is non-receptor kinase

In some cases there is more than one name of described factor and all should be listed (e.g. p21 (p21Cip1, p21Waf) is known not only as cyclin-dependent kinase inhibitory protein -1 but also cyclin-dependent kinase inhibitor 1 or CDK-interacting protein 1. The alternative names should be added to all abbreviations

Author Response

The authors of this manuscript express their sincere thanks to the reviewer for the critical assessment of this work. The authors have acted upon the recommendations of the reviewer which has resulted in a significant enhancement in the quality of this manuscript. All modifications incorporated in the manuscript are highlighted in red color font. A “point-by-point” response to each and every comment is outlined below:

General comments:

In their review Essa M. Bajalia et al. analyzed potential role of phytochemicals in RCC treatment. They described effects of more than 60 phytochemicals in both in vitro and in vivo studies. The study addressed a potentially interesting issue. Renal cell carcinoma (especially metastatic form) is therapy resistant. Knowledge about effects of natural compounds on RCC could help to develop new therapeutics. I recommend the study of Essa M. Bajalia et al. for publication in the Cancer Biomarkers after minor changes.

Response:

We thank the reviewer for his/her expertise, time, and effort for reviewing our manuscript. We are deeply encouraged by the generous comments regarding the quality of our work and the recommendation for publication. We sincerely appreciate the valuable suggestions which we have found extremely valuable while revising our manuscript.

 Specific comments:

Comment 1:

Authors often write about Caki cell line ex. Verses 245, 301, 325 etc., however there are two RCC cell lines which could be named Caki - Caki-1 and Caki-2 (Caki-1 is cell line derived from skin metastasis of ccRCC while Caki-2 is cell line derived from primary renal cell carcinoma). Authors should specify which cell line was tested in discussed articles.

Response:

We thank the reviewer for this critical comment. We agree that it is important to be able to distinguish between Caki-1 and Caki-2 cell lines. However, many of the research papers did not make this distinction. We did find two papers where this distinction could be made and have updated these changes (page 34, line 1110; page 34, line 1128). The general term Caki is used only when the authors of a research paper did not specify Caki-1 or Caki-2.

Comment 2:

In descriptions of some phytochemicals authors give information about the table in which experiments are summarized (e.g. 3.3.1 verse 1058, 1066) in others not (e.g. 3.3.8). It should be standardized - authors should add information about tables to all paragraphs or remove it.

Response:

We understand the reviewer’s concern. However, following the standard journal style, we have cited each table once where the first study of a table has been presented in the text. Since there are numerous studies summarized in each table, we feel our style is appropriate as it eliminates unnecessary frequent citations. Moreover, since the content of each table (e.g., in vitro, in vivo and clinical studies) has been clearly mentioned in the title, we feel it is appropriate for a reader to understand the synergy between text and tables.

Comment 3:

Detailed list of small mistakes which must be corrected before publication.

Response:

We thank the reviewer for carefully checking the manuscript and bringing these mistakes to our attention. We have thoroughly edited our manuscript to remove various typographical errors.

Comment 4:

3.1.16 Hymecrone

 verse 436  “Hymecrone (…) produced a downregulation” it should be changed.

Response:

We have updated the sentence to read more clearly: “Hymecromone and sorafenib in combination downregulated cluster of differentiation 44 (CD44), receptor for hyaluronan-mediated motility (RHAMM), p-MEK, p-ERK, and p-VEGFR levels” (pages 18, line 450 to and page 19, line 452).

Comment 4:

Sentence from verse 459 starting from Confirmation of apoptosis - should be corrected - it is hard to follow.

Response:

We thank the reviewer for noting this mistake. We have removed that sentence from in vivo paragraph and added the following sentence to the in vitro section of icaritin, as this was an in vitro finding:

“Additionally, icaritin increased the levels of cleaved caspase-3 and cleaved PARP, demonstrating its pro-apoptotic effect” (page 19, lines 469-471).

To further elaborate on the additional in vivo findings, we have added the following sentence:  “Furthermore, icaritin treatment reduced VEGF expression, indicating its potential for use as an antiangiogenic agent” (page 19, lines 475 and 476).

Comment 5:

3.1.202 - verse 494 - chemical name of osthole as Figure 2 are in square brackets([]) instead of normal brackets (()).

Response:

We applaud the reviewer for his/her attention to details. The use of square brackets has been necessary as the name of the compound has circular brackets. However, we have rectified the typographical error as follows: [7-methoxy-8-(3-methyl-2-butenyl) coumarin, Figure 2] (page 20, line 510).

Comment 6:

3.1.24 verses 596 and 597 p110g is not PI3K member protein - p110g is an isoform of catalytic subunit of PI3K kinase.

Response:

We agree with the reviewer and modified the sentence as follows:

“Umbelliferone demonstrated inhibition of p110γ, an isoform of catalytic subunit of PI3K, which contributes to its ability to regulate cell cycle and induce apoptosis” (page 22, lines 612 and 613).

Comment 7:

3.1.24 verse 596  - it should be PI3K instead P13K.

Response:

We have made the necessary correction (page 22, line 613).

Comment 8:

3.2.3 verse 638 there is information about two cell lines while in the Table1 3 cell lines are listed.

Response:

The reviewer is correct. We have modified the sentence as follows:

“Chauhan et al. [138] studied the efficacy of artesunate against three kidney cancer cell lines originated from papillary cell carcinoma and RCC” (page 24, lines 654 and 655).

Comment 9:

3.2.4  verses 647-648 and from 656 the same results of Jang et al. are described.

Response:

We have removed the in vivo results from the in vitro section to avoid repeating the same information. The first paragraph now contains solely the in vitro results (page 24, lines 659-668), whereas the second paragraph exclusively presents the in vivo results (page 24, lines 669-680).

Comment 10:

3.2.9 verse 748 MMP is Mitochondrial Membrane Potential (there is no such abbreviation in abbreviations list it must be corrected in text and in abbreviations list).

Response:

We now only use the abbreviation MMP for “matrix metalloproteinases” to avoid confusion with “mitochondrial membrane potential”. Changes were made to remove the abbreviation for mitochondrial membrane potential (page 26, lines 761 and 762; page 30, line 980; page 32, line 1036).

Comment 11:

3.2.11 Information about crocin effects on miR-577 expression is doubled. (verses 777and 778) the sentence form verses 778-779 should be corrected - it does not sound scientific.

Response:

We appreciate the reviewer’s comment regarding the results of crocin. To clarify the findings, we have modified the text as follows:

“It increased the expression of miR-577, which acts as a tumor suppressor. The researchers also found that nuclear factor I B (NFIB), a direct target gene of miR-577, was suppressed by crocin treatment, demonstrating its antiproliferative effect” (page 27, lines 790-793).

Comment 12:

3.3.2 sentence in verses 1082- 1083 should be corrected.

Response:

We have corrected the sentence which now states:

“When berberine and PDT were used together, there was a significant reduction in cell viability” (page 34, lines 1094 and 1095).

The results of berberine alone have been presented separately (page 33, lines 1089-1091).

Comment 13:

3.3.9 verse 1174 c-Met is not mesenchymal-epithelial factor (c-Met is HGF receptor) it should be corrected in text and in abbreviation list.

Response:

We thank the reviewer for the comment. c-Met is now changed to tyrosine protein kinase Met or hepatocyte growth factor receptor in the text (page 35, lines 1188 and1189) and removed from the abbreviation list.

Comment 14:

3.4.2 verse 1208 - The sentence from verse 1208 “Everolimus (…. ) can cause resistance (…)” must be corrected. Everolimus is a drug and during therapy patients could develop resistance for this drug; verse 1223 sulforaphane was used to prevent cells from development of resistance not to overcome it.

Response

We appreciate the reviewer’s comment and have made the following changes in the text:

“Everolimus is a chemotherapeutic drug and during treatment patients could develop resistance against this medication” (page 36, lines 1222-1224).

We have also modified the following sentence to use the word “prevent”:

“In an additional study, Juengel et al. [274] used RCC cell lines, namely Caki-1, KTCTL-26, and A498, to investigate sulforaphane’s ability to prevent resistance in everolimus-treated cells: (page 37, lines 1236-1238).

Comment 15:

Abbreviations

c-Met is not mesenchymal-epithelial factor

Response:

We have deleted c-Met from the abbreviation list and modified the text (page 35, lines 1188 and 1189).

Comment 16:

MMP are matrix metaloproteinases but in text (verse 748) MMP is also Mitochondrial Membrane Potential.

Response:

The abbreviation MMP is now only used for matrix metalloproteinases. Mitochondrial membrane potential is no longer abbreviated in the manuscript to avoid confusion between the two. Please see response to comment 10 above.

Comment 17:

SRC is non-receptor kinase.

Response:

We have modified the text (page 32, line 1071) and removed it from the abbreviation list.

Comment 18:

In some cases there is more than one name of described factor and all should be listed (e.g. p21 (p21Cip1, p21Waf) is known not only as cyclin-dependent kinase inhibitory protein -1 but also cyclin-dependent kinase inhibitor 1 or CDK-interacting protein 1. The alternative names should be added to all abbreviations.

Response:

We have deleted p21 from the abbreviation list and modified the text to include the alternative names for p21 (page 17, lines 351 and 352).

Additionally,

  1. The entire manuscript has been thoroughly checked and edited to minimize typographical errors as well as to ensure uniform style, organization, and quality.
  2. The reference list has been modified as we have added several new references. Special attention is given to conform to the order of references and bibliographic style of the journal.

Finally,

On behalf of my co-authors, I once again express my sincere thanks to the erudite reviewer for the valuable suggestions and constructive input to improve the quality of our manuscript.

Reviewer 2 Report

    The authors reviewed the current developments in phytochemicals for the prevention and treatment of renal cell carcinoma from a preclinical and clinical standpoints and uncovered some molecular mechanisms.

    Points to be considered:

    1)the rationale of why the authors came up with this review.

    2) what is the information that is not exactly available that motivated the authors to come up with this information. what are the current caveats and how do the authors highlight the current research in answering them? if not they need to address in future directions.

    3) the authors need to highlight what new information the review is providing to enhance the research in progress.

    4) in the discussion, this reviewer personally misses some insights regarding novel concepts in RCC: as is now well known, tumors grow and evolve through constant crosstalk with the surrounding microenvironment, and emerging evidence indicates that angiogenesis and immunosuppression frequently occur simultaneously in response to this crosstalk. accordingly, strategies combining anti-angiogenic therapy and immunotherapy seem to have the potential to tip the balance of the tumor microenvironment and improve treatment response. prostate carcinoma does not make an exception (please refer to PMID: 32456352  and PMID: 33791209 and expand).

    5) Indeed, a graphical abstract and a workflow with the authors bullets in this regard (point 4) would boost the interest for a broad readership

    Author Response

    The authors of this manuscript express their sincere thanks to the reviewer for the critical assessment of this work. The authors have acted upon the recommendations of the reviewer which has resulted in a significant enhancement in the quality of this manuscript. All modifications incorporated in the manuscript are highlighted in red color font. A “point-by-point” response to each and every comment is outlined below:

    General comments:

    The authors reviewed the current developments in phytochemicals for the prevention and treatment of renal cell carcinoma from a preclinical and clinical standpoints and uncovered some molecular mechanisms.

    Response:

    We thank the reviewer for his/her expertise, time, and effort for reviewing our manuscript. We are deeply encouraged by the generous comments on our work. We sincerely appreciate the valuable suggestions which we have found extremely valuable while revising our manuscript.

    Specific comments:

    Comment 1:

    The rationale of why the authors came up with this review.

    Response:

    We thank the reviewer for the comment. We believe the objectives and novelty of our review have been highlighted in the last paragraph of the Introduction section. However, we have added the following to the Introduction section to expand on the rationale as to why we came up with this review:

    “With RCC being an aggressive and deadly cancer, there is great potential for the use of phytochemicals as drug therapy. A comprehensive review of phytochemicals and their anticancer effects will further consolidate information and serve as a reference for future researchers targeting novel drug therapy” (page 3, lines 127-130).

    We also feel this rationale is emphasized in our Conclusion section which states:

    “It is our hope that the information provided in this work would be useful for those exploring therapeutic avenues of RCC by creating new treatment protocol with minimal side effects” (page 43, lines 1457-1459).

    Comment 2:

    What is the information that is not exactly available that motivated the authors to come up with this information. what are the current caveats and how do the authors highlight the current research in answering them? if not they need to address in future directions.

    Response:

    We appreciate the reviewer’s comment regarding the motivation and current caveats of the current research. We have stated our motivation, as well as the current limitations to the available research in the Introduction section (page 3, lines 114-134).

    To elaborate on these points, we have added the following text to the Conclusion section:

    “Previous reviews of RCC and phytochemical therapy involved fewer phytochemicals, did not include mechanisms of actions, or did not include all types of studies, such as in vitro, in vivo, and clinical trials. This review encompasses a complete review of phytochemicals with all study types for potential use in the treatment of RCC” (page 41, lines 1369-1373).

    Comment 3:

    The authors need to highlight what new information the review is providing to enhance the research in progress.

    Response:

    To highlight the new information brought about by this review, specifically its ability to enhance the current research in progress, we have added the following text:

    “With comprehensive knowledge of these critical molecular targets and pathways, this information can further direct researchers as to which targets affect oxidative stress, inflammation, proliferation, cell cycle arrest, cell death, angiogenesis, and metastasis. Identifying these targets is crucial, considering targeted therapy may provide improved outcomes in the treatment of RCC” (page 42, lines 1395-1399).

    Comment 4:

    In the discussion, this reviewer personally misses some insights regarding novel concepts in RCC: as is now well known, tumors grow and evolve through constant crosstalk with the surrounding microenvironment, and emerging evidence indicates that angiogenesis and immunosuppression frequently occur simultaneously in response to this crosstalk. accordingly, strategies combining anti-angiogenic therapy and immunotherapy seem to have the potential to tip the balance of the tumor microenvironment and improve treatment response. prostate carcinoma does not make an exception (please refer to PMID: 32456352  and PMID: 33791209 and expand).

    Response:

    We thank the reviewer for bringing up this very important point regarding the crosstalk between tumors and their surrounding environment, and the role this plays in targeted therapy. We have added the following text to our revised manuscript:

    “The compilation of these various phytochemical mechanisms will further allow researchers to target RCC therapy by accounting for cell line heterogeneity and modulating tumor microenvironment. Marquardt et. al. [290] studied the different histopathological groups of RCC cells lines and found that a substantial number of samples diverge from the RCC RNAseq data. These subgroups were found to be characterized by strengthened mitochondrial and weakened angiogenesis gene signatures. This indicates the importance of identifying the genetically divergent RCC subgroups and developing specific drug therapy for these various genotypes. Multiple potential drug targets for RCC have been investigated for tailored treatment in a study by Argentiero et al. [291]. With the knowledge of subgroups of RCC cell lines with differing histopathological and genetic composition and thus unique tumor microenvironments, the combination therapy of anti-angiogenic and immunomodulatory targets is promising. Therefore, identifying a multitude of anticancer drug targets via phytochemicals can potentially provide a more individualized approach to RCC treatment” (page 42, lines 1406-1419).

    Comment 5:

    Indeed, a graphical abstract and a workflow with the authors bullets in this regard (point 4) would boost the interest for a broad readership.

    Response:

    We again thank the reviewer for the comment. We have added a graphical abstract to the manuscript which we believe will increase the interest for a broader readership as suggested. This graphical abstract summarizes the various molecular targets of phytochemicals in RCC.

    Additionally,

    1. The entire manuscript has been thoroughly checked and edited to minimize typographical errors as well as to ensure uniform style, organization, and quality.
    2. The reference list has been modified as we have added several new references. Special attention is given to conform to the order of references and bibliographic style of the journal.

    Finally,

    On behalf of my co-authors, I once again express my sincere thanks to the erudite reviewer for the valuable suggestions and constructive input to improve the quality of our manuscript.

    Round 2

    Reviewer 2 Report

    The authors have clarified several of the questions I raised in my previous review. Most of the major problems have been addressed by this revision.